# Addressing Mark Imbalance in Integration-free Neural Marked Temporal Point Processes

**Sishun Liu**
RMIT University
Melbourne, Victoria 3000
sishun.liu@student.rmit.edu.au

**Ke Deng**
RMIT University
Melbourne, Victoria 3000
ke.deng@rmit.edu.au

**Yongli Ren**
RMIT University
Melbourne, Victoria 3000
yongli.ren@rmit.edu.au

**Yan Wang**
Macquarie University
Syndey, New South Wales 2000
yan.wang@mq.edu.au

**Xiuzhen Zhang**
RMIT University
Melbourne, Victoria 3000
xiuzhen.zhang@rmit.edu.au

## Abstract

Marked Temporal Point Process (MTPP) has been well studied to model the event distribution in marked event streams, which can be used to predict the mark and arrival time of the next event. However, existing studies overlook that the distribution of event marks is highly imbalanced in many real-world applications, with some marks being frequent but others rare. The imbalance poses a significant challenge to the performance of the next event prediction, especially for events of rare marks. To address this issue, we propose a *thresholding* method, which learns thresholds to tune the mark probability normalized by the mark's prior probability to optimize mark prediction, rather than predicting the mark directly based on the mark probability as in existing studies. In conjunction with this method, we predict the mark first and then the time. In particular, we develop a novel neural MTPP model to support effective time sampling and estimation of mark probability without computationally expensive numerical improper integration. Extensive experiments on real-world datasets demonstrate the superior performance of our solution against various baselines for the next event mark and time prediction. The code is available at https://github.com/undes1red/IFNMTPP.

## 1 Introduction

*Marked Temporal Point Process (MTPP)* models event sequences observed from natural phenomena (e.g. earthquakes) or generated in human activities (e.g. retweets), where each event has a mark and an arrival time. MTPP has attracted the attention of the research community (see Shchur et al. [34] for a comprehensive review). Typically, MTPP models the joint *Probability Distribution Function (PDF)* conditioned on history, denoted as $p^*(m,t)$[1], where $m$ and $t$ are the mark and arrival time[2] of the next event, respectively. Some studies are on the Poisson Process [37] and Hawkes process

---

[1]The asterisk reminds the probability is conditioned on history, i.e., the events in the past.
[2]The time relative to the most recent event

39th Conference on Neural Information Processing Systems (NeurIPS 2025).

[32, 16, 27]. Recently, we have witnessed a rapid growth of *neural MTPP*, which models $p^*(m, t)$ using neural networks [24, 28, 41, 26, 43], due to the capability of learning complicated temporal patterns and computational efficiency [34].

However, existing studies overlook that the distribution of event marks is highly imbalanced in many real-world applications, with some marks being frequent but others rare, as shown in Figure 1 (a). Similar to other machine learning tasks such as classification, the imbalance poses a significant challenge to the performance of the next event prediction, especially for events of rare marks, which are often more important than other marks (e.g., the occurrence of a 7-magnitude earthquake or a retweet from celebrities). By mitigating the impact of mark imbalance, this study aims to improve the performance of MTPP for next event prediction.

Various techniques have been investigated to improve the prediction performance of rare classes in classifiers, including *resampling the training set*, *cost-sensitive approaches*, and *thresholding* [17, 1, 39]. Training data resampling requires a proper resampling ratio. Cost-sensitive approaches require domain knowledge on the importance of different marks in setting the cost [17]. To have a solution with minimum external knowledge and assumptions, this study adopts thresholding, which learns thresholds to tune the mark probability normalized by the prior probability of marks.

Addressing mark imbalance for MTPP using thresholding is not straightforward. In addition to mark prediction, MTPP also needs to predict the time simultaneously. In most existing MTPP studies, the strategy is to predict the time based on $p^*(t)$, the probability that the next event time is $t$, and then predict the mark based on $p^*(m|t)$, the probability that the next event mark is $m$ at the predicted time $t$. Our analysis and experiments show that this strategy is unsuitable for addressing mark imbalance with thresholding. If time changes , the mark probability conditioned on time typically changes and thus requires different tuning thresholds. However, it is implausible to learn the tuning thresholds at all times. So, we propose a strategy that first predicts the mark based on $p^*(m)$, the probability that the next event mark is $m$, and then predicts the time based on $p^*(t|m)$, the probability that the next event time is $t$ on the condition that the predicted mark $m$ is the next event mark. Since the mark probability $p^*(m)$ is independent of time, applying thresholding to handle mark imbalance is easy.

However, our strategy has its challenges. First, two different improper integrations are required for modeling $p^*(m)$ and time prediction, respectively. Second, sampling $p^*(t|m)$ to predict time is inefficient because it needs the *Cumulative Distribution Function (CDF)* of $p^*(t|m)$, but the CDF does not have a closed-form expression. To overcome these challenges, we find a way to unify the two improper integrations into one. Then, we develop a novel MTPP model, called *Integration-free Neural Marked Temporal Point Process (IFNMTPP)*, to approximate the unified improper integration, rather than using a computationally expensive numerical method. With IFNMTPP, we can directly model $p^*(m)$ and the CDF of $p^*(t|m)$. The CDF makes drawing samples from $p^*(t|m)$ efficient for time prediction. Based on $p^*(m)$, the thresholding method can be applied to address the mark imbalance. Extensive experiments on real-world datasets demonstrate the superior performance of our solution against various baselines for the next event mark and time prediction. The contributions of this study are threefold:

- This study investigates the impact of mark imbalance on MTPP for next event prediction, which is overlooked by existing MTPP studies.
- This study introduces the first solution to address mark imbalance in MTPP, which learns thresholds to tune the mark probability normalized by the prior probability of marks to optimize mark prediction, rather than predicting marks based on mark probability directly as in existing studies.
- This study finds a way to unify two improper integrations into one, and proposes a novel Integration-free Neural Marked Temporal Point Process (IFNMTPP) to approximate the unified improper integration to support time sampling and estimation of mark probability, rather than using computationally expensive numerical improper integration.

## 2 Preliminaries

### 2.1 Marked Temporal Point Process

The Marked Temporal Point Process (MTPP) is a random process whose embodiment is a sequence of discrete events, $\mathcal{S} = \{(m_i, t_i)\}_{i=1}$, where $i \in \mathbb{Z}^+$ is the sequence order, $t_i \in \mathbb{R}^+$ is the time

when the $i$th event occurs, $m_i$ is the mark of the $i$th event. This study only concerns a finite set of categorical marks $M = \{k_1, k_2, \cdots, k_{|M|}\}$, and the simple MTPP, which allows at most one event at every time, thus $t_i < t_j$ if $i < j$. The time of the most recent event is $t_l$, and the current time is $t > t_l$. The time interval between two adjacent events is the inter-event time. We assume that an event with a particular mark at a particular time may be triggered by past events. Let $\mathcal{H}_{t_l}$ be the history up to (including) the most recent event, and $\mathcal{H}_{t-}$ be the history up to (excluding) the current time [31]. With these definitions, we can define the *Conditional Intensity Function (CIF)* of MTPP:

$$\lambda^*(m = k_i, t) = \lambda(m = k_i, t | \mathcal{H}_{t-}) = \lim_{\Delta t \to 0} \frac{P(m = k_i, t \in [t, t + \Delta t] | \mathcal{H}_{t-})}{\Delta t}. \quad (1)$$

With $\lambda^*(m, t)$, the conditional joint PDF of the next event can be defined:

$$p^*(m, t) = p(m, t | \mathcal{H}_{t_l}) = \lambda^*(m, t) F^*(t) = \lambda^*(m, t) \exp(-\int_{t_l}^t \sum_{n \in M} \lambda^*(n, \tau) d\tau). \quad (2)$$

where $\tau$ means integrating over time. $F^*(t)$ is the conditional PDF that no event has ever happened up to time $t$ since $t_l$. We explain how to obtain Equation (2) from Equation (1) in Appendix A.

The simplest form of MTPP is the homogeneous Poisson process whose CIF merely contains a positive number, i.e., $\lambda^*(m = k_i, t) = c$. Another example is the Hawkes process [14], belonging to the self-exciting point process family. Its CIF is $\lambda^*(m = k_i, t) = \mu_i + \sum_{j:t_j < t} \kappa_i(t, t_j)$ where $\kappa_i(t, t_j) > 0$ represents the excite from previous events. Because it meets the real-world intuition that the influence of occurred events always drastically drops as time passes, the Hawkes process is a widely used backbone process in various models [4, 27, 16, 15]. Recently, we have witnessed a rapid growth of neural MTPP, which models $p^*(m, t)$ using neural networks [24, 28, 41, 26, 43], due to the capability of learning complicated temporal patterns and computational efficiency [34].

Based on $p^*(m, t)$, the mark $m$ and time $t$ of the next event can be predicted. Most existing MTPP methods predict when the next event will occur first, and then predict what the mark is at the predicted time. Specifically, the expected time of the next event is $\bar{t} = \int_{t=t_l}^{\infty} \tau p^*(\tau) d\tau$ where $p^*(t) = \sum_{m \in M} p^*(m, t)$. A numerical method is typically used to calculate $\bar{t}$ by sampling $N$ times, denoted as $\{t^i\}_N$, from $p^*(t)$ following *Thinning Algorithm (TA)* or *Inverse Transform Sampling (ITS)* [31] so that $\bar{t} = \frac{1}{N} \sum_i t^i$. After that, the mark of the next event at $\bar{t}$ is predicted: $m_{\bar{t}} = \arg\max_{m \in M} p^*(m, \bar{t})$. Some studies predict the mark of the next event $m = \arg\max_{m \in M} p^*(m)$ and then predict the time of the next event $\bar{t}_m = \int_{t=t_l}^{\infty} \tau p^*(\tau | m) d\tau$ given the predicted mark [36].

## 2.2 Mark Imbalance

In real-world scenarios, the mark distribution can be significantly imbalanced, i.e., some marks are persistent and others are rare. The imbalance hurts the performance of the next event prediction, especially for rare marks, which are often more important than other marks (e.g., the occurrence of a 7-magnitude earthquake). Let us consider two marks $k_1$ and $k_2$ where $k_1$ is much more frequent than $k_2$ in the observed event sequence. Suppose the next event is mark $k_2$. Because $k_1$ is much more frequent than $k_2$, it is very likely that $p^*(k_1, t) > p^*(k_2, t)$ for most of the time $t$, including $\bar{t} = \int_{t=t_l}^{\infty} \tau \sum_{m \in M} p^*(m, \tau) d\tau$. If so, $k_1$ will be predicted as the next event, but the real mark is $k_2$.

Figure 1 (a) demonstrates the mark frequency distribution in three datasets, Retweet, USearthquake, and StackOverflow (see details in Section 4). Figure 1 (b) shows $p^*(m, t)$ for each mark $m$ in these datasets. The envelope covers $p^*(m, t)$ of all instances in the datasets, and the line is the average of $p^*(m, t)$ across these instances. These figures show that $p^*(k_1, t) > p^*(k_2, t)$ for most of the time if $k_1$ is frequent and $k_2$ is rare. Table 1 shows the mark prediction performance achieved by SAHP [41], a neural MTPP model, on rare and frequent marks, respectively, measured by macro-F1. We can see that the prediction performance for rare marks is significantly lower than that for frequent marks.

## 3 Methodology

To improve the performance of MTPP for next event prediction, this study handles mark imbalance with a thresholding method, which learns thresholds to tune the mark probability normalized by the

Table 1: Mark prediction performance measured by macro-F1 using SAHP [41] for rare and frequent marks on three real-world datasets.

| | | |
|---|---|---|
| Retweet | Rare Marks | $0.0266_{\pm 0.0135}$ |
| | Freq Marks | $0.6183_{\pm 0.0010}$ |
| USearthquake | Rare Marks | $0.0037_{\pm 0.0010}$ |
| | Freq Marks | $0.2196_{\pm 0.0016}$ |
| StackOverflow | Rare Marks | $0.0863_{\pm 0.0032}$ |
| | Freq Marks | $0.2054_{\pm 0.0011}$ |

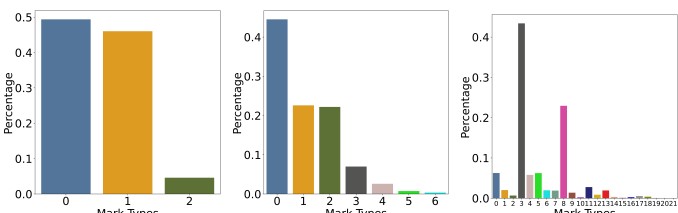

(a) The frequency distribution of marks in Retweet, USearthquake, and StackOverflow (from left to right).

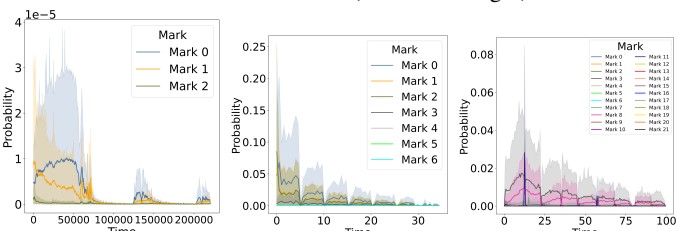

(b) The $p^*(m, t)$ for each mark $m$ in Retweet, USearthquake, and StackOverflow (from left to right).

Figure 1: A demonstration of the mark imbalance in various datasets and its influence on $p^*(m, t)$.

mark's prior probability to optimize mark prediction. In conjunction with the thresholding method, the proposed method predicts the mark based on $p^*(m)$ and then, given the predicted mark $m$, predicts the time based on $p^*(t|m)$.

## 3.1 Next Event Mark Prediction with Thresholding

The mark prediction of the next event depends on accurately modeling $p^*(m)$ for each mark $m$, the probability that the mark of the next event is $m$, based on $p^*(m, t)$. The expression of $p^*(m)$ is:

$$p^*(m) = \int_{t_l}^{+\infty} p^*(m, \tau) d\tau \tag{3}$$

In general, the frequent mark has a high $p^*(m)$ and the rare mark has a low $p^*(m)$. Inspired by the thresholding method [19, 7], we normalize the probability for each mark by its prior probability and learn to tune it to improve the prediction performance for rare marks. Specifically, for mark $m$, we calculate the ratio between the probability of $m$, $p^*(m)$, and its prior probability, $\overline{p}^*(m)$:

$$r_m = \frac{p^*(m)}{\overline{p}^*(m)} \tag{4}$$

In this paper, $\overline{p}^*(m)$ is the proportion of mark $m$ in the training set. $p^*(m)$ measures the probability that the next event mark is $m$. If $m$ is more frequent than $m'$, $p^*(m)$ is expected to be higher than $p^*(m')$. In contrast, $r_m$ evaluates whether $p^*(m)$ is higher relative to its own proportion. A rare mark having a low $p^*(m)$ may have a high $r_m$ to signal a high chance of being the next event mark. In this way, the chance of rare marks in the next event prediction is rectified.

With $r_m$ for every mark $m$, the next event mark prediction $m_p$ is obtained by a thresholding method:

$$m_p = \arg\max_{m} (r_m - \epsilon_m) \tag{5}$$

where $\epsilon_m$ is the threshold of $r_m$. We learn $\epsilon_m$, which maximizes the accuracy of $m_p$ on the training set. Specifically, for each mark $m$, if we predict the next mark as $m$ when $r_m > \epsilon_m$ and not $m$ when $r_m \leqslant \epsilon_m$, the learned $\epsilon_m$ maximizes the F1 score of the pairwise comparison, i.e., mark $m$ vs. all other marks. The technical details of the thresholding method are in Appendix C.2.

## 3.2 Next Event Time Prediction

After mark prediction, we predict the time. Let $p^*(t|m)$ be the PDF of the next event time on the condition that the next event mark is $m$. Based on $p^*(t|m)$, we have:

$$\bar{t}_m = \mathbb{E}_{t \sim p^*(\tau|m)}[t] = \int_{t_l}^{+\infty} \tau p^*(\tau|m) d\tau \tag{6}$$

where $\bar{t}_m$ is the expected time of the next event given the mark $m$.

## 3.3 Unifying Integral Functions

By the definition in Equation (3) and Equation (6), we must solve the improper integration of $p^*(m, \tau)$ and $\tau p^*(\tau|m)$ for mark probability $p^*(m)$ and time prediction $\bar{t}$, respectively. In general, improper integration does not have analytic solutions. This means that directly calculating $p^*(m)$ and $\bar{t}_m$ following Equation (3) and Equation (6) is impossible. The solution is to approximate $p^*(m)$ and $\bar{t}$. In particular, $\bar{t}$ is approximated as the average of $N$ samples $\{t^i\}_N^m$ from $p^*(t|m)$ as Equation (7).

$$\bar{t}_m = \mathbb{E}_{t \sim p^*(\tau|m)}[t] \approx \frac{1}{N} \sum_{i=1}^{N} t^i \tag{7}$$

To draw $\{t^i\}_N^m$ from $p^*(t|m)$, we use Inverse Transform Sampling (ITS), which takes the Cumulative Distribution Function (CDF) of the distribution that one wants to sample from. In our case, let $F^*(t|m)$ be the CDF of $p^*(t|m)$, i.e., $F^*(t|m) = \int_{t_l}^{t} p^*(\tau|m) d\tau$. $F^*(t|m)$ refers to the probability of the next event happening in $(t_l, t]$ on the condition that its mark is $m$. To draw a sample $t^i$ from $p^*(t|m)$, we need to solve Equation (8).

$$F^*(t^i|m) = u^i \tag{8}$$

where $u^i$ is a random sample from a uniform distribution $U(0, I)$. Since $F^*(t|m)$ is monotonic, Equation (8) is solvable by the bisection method. For each mark $m$, we obtain $\{t^i\}_N^m$ by solving Equation (8) $N$ times, which allows acquiring an arbitrary number of samples for time prediction no matter rare or frequent the mark $m$ is. We can express $F^*(t|m)$ as follows:

$$F^*(t|m) = \frac{F^*(m, t)}{p^*(m)} = \frac{1}{\int_{t_l}^{+\infty} p^*(m, \tau) d\tau} \int_{t_l}^{t} p^*(m, \tau) d\tau \tag{9}$$

where $p^*(m) = \int_{t_l}^{+\infty} p^*(m, \tau) d\tau$ is the probability that the mark of next event is $m$ since $t_l$, and $F^*(m, t) = \int_{t_l}^{t} p^*(m, \tau) d\tau$ is the probability that the next event is mark $m$ and happens in time interval $(t_l, t]$. We can further breakdown $F^*(m, t)$ as shown in Equation (10).

$$F^*(m, t) = \int_{t_l}^{t} p^*(m, \tau) d\tau = \int_{t_l}^{+\infty} p^*(m, \tau) d\tau - \int_{t}^{+\infty} p^*(m, \tau) d\tau \tag{10}$$

For each mark $m \in \mathrm{M}$, we define $\Gamma^*(m, t)$ as the integration starting from time $t$, any time after $t_l$ or $t_l$, to positive infinity:

$$\Gamma^*(m, t) = \int_{t}^{+\infty} p^*(m, \tau) d\tau \tag{11}$$

$\Gamma^*(m, t)$ is monotonically decreasing as its derivative $-p^*(m, t)$ is always smaller than 0. We rewrite $p^*(m)$ in Equation (3) and $F^*(t|m)$ in Equation (9) using $\Gamma^*(m, t)$:

$$p^*(m) = \Gamma^*(m, t_l) \tag{12}$$

$$F^*(t|m) = \frac{\Gamma^*(m, t_l) - \Gamma^*(m, t)}{\Gamma^*(m, t_l)} \tag{13}$$

This means two improper integrations in Equation (3) and Equation (6) are now unified into one, i.e., $\Gamma^*(m, t)$, for modeling $p^*(m)$ and time prediction.

While drawing samples from a distribution can follow Thinning Algorithm (TA) or Inverse Transform Sampling (ITS) [31], only ITS is suitable for integral function unification here. The basic idea in ITS is to simulate using CDF of $p^*(t|m)$. Instead, Thinning Algorithm (TA) explicitly requires the expression of $p^*(t|m)$, which is unknown typically.

## 3.4 Integration-free Neural Marked Temporal Point Process (IFNMTPP)

With $\Gamma^*(m,t)$, we can model $p^*(m)$ and prediction time $\bar{t}_m$. However, $\Gamma^*(m,t)$ is an improper integral with an infinite integration interval. Numerical methods are computationally expensive and can only be used to estimate integrals on a finite interval. To avoid numerical methods, we introduce Integration-free Neural Marked Temporal Point Process (IFNMTPP) to approximate $\Gamma^*(m,t)$. For each mark $m$, IFNMTPP models the relationship between $p^*(m,t)$ and its integral $\Gamma^*(m,t)$.

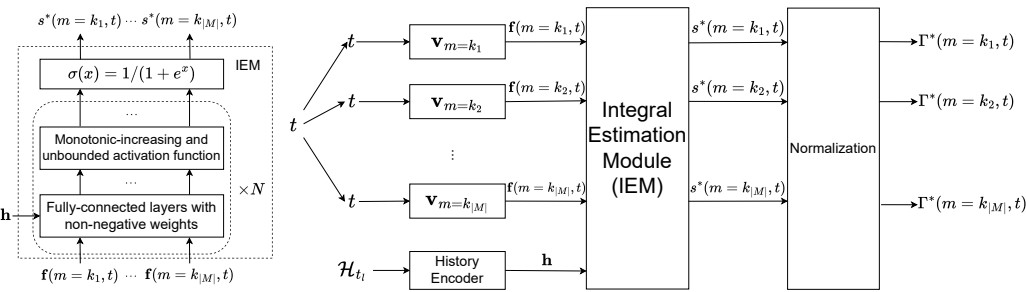

Figure 2: Architecture of IFNMTPP where the history encoder is an LSTM.

Figure 2 sketches the architecture of IFNMTPP. For each mark $m \in \mathrm{M}$, we assign a vector $\mathbf{v}_m$ to prepare $\mathbf{f}(m,t) = \mathbf{v}_m(t - t_l) + \mathbf{b}_m$ as input of the Integral Estimation Module (IEM). All parameters in $\mathbf{v}_m$ are non-negative. IEM contains multiple fully-connected layers with non-negative weights, and monotonic-increasing and unbounded activation functions. Then, it ends with a monotonically decreasing function $\sigma(x) = 1/(1+e^x)$. So, IFNMTPP is intrinsically monotonically decreasing w.r.t. $t$. The outputs of IEM are scores $s^*(m = k_1, t), s^*(m = k_2, t), \cdots, s^*(m = k_{|\mathrm{M}|}, t)$. The value of $\sum_{m \in \mathrm{M}} s^*(m,t)$ is not guaranteed to be 1. To produce the qualified probability distribution, they need to be normalized. This is achieved by the Normalization module in Figure 2 that divides $s^*(m,t)$ by the partition function $Z(\mathcal{H}_{t_l}) = \sum_{m \in \mathrm{M}} s^*(m, t_l)$ for each $m \in M$. Finally, IFNMTPP outputs $\Gamma^*(m,t)$ for each mark $m$ at the given time $t$:

$$\Gamma^*(m,t) = \frac{s^*(m,t)}{Z(\mathcal{H}_{t_l})} \tag{14}$$

With $\Gamma^*(m,t)$ and $\Gamma^*(m,t_l)$, we have $F^*(t|m)$ by Equation (9) and Equation (10). Next, we calculate $\bar{t}_m$ by drawing $\{t^i\}_N^m$ from $F^*(t|m)$ following Equation (8). With the definition of IFNMTPP, we have the following proposition, with the proof in Appendix B, to guarantee that the model output is $\Gamma^*(m,t)$:

**Proposition 3.1.** *The output of IFNMTPP is* $\Gamma^*(m,t)$ *when its gradient is* $-p^*(m,t)$.

We train IFNMTPP using the Negative Log-Likelihood (NLL) on event sequence $\mathcal{S}$ observed in a time interval $[t_0, T)$, where the time of the first event is $t_1 \geq t_0$, and the time of the last event is $t_s \leq T$.

$$
\begin{aligned}
L = -\log p(\mathcal{S}) &= -\sum_{(m_i, t_i) \in \mathcal{S}} \log \lambda^*(m_i, t_i) + \int_{t_0}^{T} \sum_{n \in M} \lambda^*(n, \tau) d\tau \\
&= -\sum_{(m_i, t_i) \in \mathcal{S}} \left( \log \lambda^*(m_i, t_i) - \int_{t_{i-1}}^{t_i} \sum_{n \in M} \lambda^*(n, \tau) d\tau \right) + \int_{t_s}^{T} \sum_{n \in M} \lambda^*(n, \tau) d\tau \\
&= -\sum_{(m_i, t_i) \in \mathcal{S}} \log p^*(m_i, t_i) - \log(1 - \sum_{n \in M} F^*(n, T)) \\
&= -\sum_{(m_i, t_i) \in \mathcal{S}} \log p^*(m_i, t_i) - \log(\sum_{n \in M} \Gamma^*(n, T))
\end{aligned}
\tag{15}
$$

where $p^*(m_i, t_i)$ is the probability of the $i$th event conditioned on historical events. IFN-MTPP increases $p^*(m_i, t_i)$ where $(m_i, t_i) \in \mathcal{S}$ are real events in event sequences. The term

Table 2: Mark prediction performance measured by macro-F1/micro-F1 to evaluate thresholding and the prediction order of mark and time. The bold are the best values.

| | | Retweet | SO | Taobao | USearthquake |
|---|---|---|---|---|---|
| ours | M | **$0.4750_{\pm0.0033}$/$0.4394_{\pm0.0093}$** | **$0.1776_{\pm0.0030}$/$0.6376_{\pm0.0026}$** | **$0.4190_{\pm0.0104}$/$0.7499_{\pm0.0151}$** | **$0.1382_{\pm0.0071}$/$0.3189_{\pm0.0125}$** |
| | $M_r$ | **$0.2010_{\pm0.0082}$/$0.2010_{\pm0.0082}$** | **$0.1476_{\pm0.0041}$/$0.4530_{\pm0.0026}$** | **$0.3987_{\pm0.0108}$/$0.7558_{\pm0.0185}$** | **$0.0339_{\pm0.0051}$/$0.1111_{\pm0.0098}$** |
| | $M_f$ | **$0.6120_{\pm0.0013}$**/$0.9612_{\pm0.0021}$ | $0.2795_{\pm0.0014}$/$0.8974_{\pm0.0042}$ | **$0.7441_{\pm0.0060}$/$0.7441_{\pm0.0060}$** | **$0.2773_{\pm0.0215}$**/$0.9181_{\pm0.0102}$ |
| time-mark-with-thresholding | M | $0.4741_{\pm0.0028}$/$0.4380_{\pm0.0016}$ | $0.1431_{\pm0.0075}$/$0.5834_{\pm0.0028}$ | $0.3289_{\pm0.0193}$/$0.7059_{\pm0.0384}$ | $0.1214_{\pm0.0126}$/$0.3091_{\pm0.0083}$ |
| | $M_r$ | $0.2000_{\pm0.0067}$/$0.2000_{\pm0.0011}$ | $0.1023_{\pm0.0094}$/$0.3829_{\pm0.0046}$ | $0.3054_{\pm0.0201}$/$0.7072_{\pm0.0690}$ | $0.0298_{\pm0.0093}$/$0.1049_{\pm0.0059}$ |
| | $M_f$ | $0.6093_{\pm0.0008}$/$0.9596_{\pm0.0019}$ | **$0.2815_{\pm0.0022}$**/$0.8888_{\pm0.0041}$ | $0.7062_{\pm0.0204}$/$0.7062_{\pm0.0204}$ | $0.2436_{\pm0.0236}$/$0.9116_{\pm0.0064}$ |
| ours-w/o-thresholding | M | $0.4269_{\pm0.0010}$/$0.1800_{\pm0.0093}$ | $0.1287_{\pm0.0031}$/$0.5877_{\pm0.0023}$ | $0.3968_{\pm0.0138}$/$0.7183_{\pm0.0220}$ | $0.1153_{\pm0.0061}$/$0.1172_{\pm0.0048}$ |
| | $M_r$ | $0.0333_{\pm0.0082}$/$0.0333_{\pm0.0034}$ | $0.1065_{\pm0.0047}$/$0.3763_{\pm0.0030}$ | $0.3759_{\pm0.0150}$/$0.7059_{\pm0.0332}$ | $0.0121_{\pm0.0035}$/$0.0146_{\pm0.0012}$ |
| | $M_f$ | $0.6238_{\pm0.0001}$/**$0.9770_{\pm0.0000}$** | $0.2043_{\pm0.0022}$/**$0.9180_{\pm0.0004}$** | $0.7311_{\pm0.0108}$/$0.7311_{\pm0.0108}$ | $0.2528_{\pm0.0118}$/**$0.9451_{\pm0.0005}$** |
| time-mark-w/o-thresholding | M | $0.4252_{\pm0.0033}$/$0.1815_{\pm0.0077}$ | $0.0906_{\pm0.0055}$/$0.4428_{\pm0.0080}$ | $0.3135_{\pm0.0136}$/$0.6384_{\pm0.0316}$ | $0.1066_{\pm0.0040}$/$0.1111_{\pm0.0336}$ |
| | $M_r$ | $0.0338_{\pm0.0029}$/$0.0338_{\pm0.0029}$ | $0.0567_{\pm0.0071}$/$0.2142_{\pm0.0076}$ | $0.2897_{\pm0.0137}$/$0.5884_{\pm0.0467}$ | $0.0033_{\pm0.0024}$/$0.0143_{\pm0.0092}$ |
| | $M_f$ | $0.6208_{\pm0.0008}$/**$0.9770_{\pm0.0001}$** | $0.2059_{\pm0.0004}$/$0.9156_{\pm0.0013}$ | $0.6933_{\pm0.0129}$/$0.6933_{\pm0.0129}$ | $0.2444_{\pm0.0085}$/$0.9446_{\pm0.0002}$ |

$\log(\sum_{m \in M} \Gamma^*(m, T))$ is the survival term, which models no events after the last event $t_s$ in each event sequence until time $T$. In IFNMTPP, the expression of $p^*(m_i, t_i)$ is:

$$p^*(m_i, t_i) = -\frac{\partial \Gamma^*(m_i, t_i)}{\partial t_i} = -\frac{1}{Z(\mathcal{H}_{t_l})} \frac{\partial s^*(m_i, t_i)}{\partial \mathbf{f}(m_i, t_i)} \frac{\partial \mathbf{f}(m_i, t_i)}{\partial t_i} \tag{16}$$

Mei et al. [25] prove that an MTPP model converges to the true distribution when trained with the NLL loss defined in Equation (15). Combined with Proposition 3.1, IFNMTPP consistently estimates the true value of $\Gamma^*(m, t)$.

## 4 Experiments

We run every experiment 5 times with different random seeds and report the mean and standard deviation (1-sigma) of all results. The complete experiment settings are described in Appendix D.

**Datasets**[3] Four real-world datasets include Retweet [42], StackOverflow(SO) [20], Taobao User Behavior Data(Taobao) [2], and earthquake events over the Conterminous US(USearthquake) [38]. We split all marks of each dataset into two subsets, one containing frequent marks, denoted as $M_f$, the other containing rare marks, denoted as $M_r$. $M_r \cap M_f = \varnothing$ and $M_r \cup M_f = M$. The rare marks and frequent marks for each dataset are described in Appendix D.5.

**Baseline Models**[4] Our method, denoted as *ours*, uses IFNMTPP for predicting the mark of the next event, optimized with thresholding, then uses IFNMTPP to predict the time of the next event given the predicted mark. The first group of baselines includes: (i) *ours-w/o-thresholding* to evaluate the effectiveness of the thresholding method. (ii) *time-mark-with-thresholding* to evaluate the necessity to predict marks first for handling mark imbalance with thresholding. (iii) *time-mark-w/o-thresholding* same as time-mark-with-thresholding but mark prediction is not optimized with thresholding. The second group of baselines evaluates thresholding against resampling, another classic technique to address data imbalance, including *undersampling* and *oversampling*. The third group of baselines includes existing MTPP methods. Since MTPP modeling has been well studied in the past decades, the state-of-the-art methods demonstrate comparable performance. Among them, this study selects the most popular ones as baselines, including FullyNN [28], THP [44], SAHP [41], AttNHP [26], and Marked-LNM [36]. The details of these baselines are available in Appendix D.4.

**Evaluation Metrics** We use macro-F1 and micro-F1, described in Appendix D.3.2, to evaluate mark predictions and use Mean Absolute Error (MAE), described in Appendix D.3.3, to evaluate time predictions on real-world datasets. The evaluation metrics for mark and time are independent of each other. For each dataset, as discussed above, three sets $M$, $M_f$, and $M_r$ are drawn from the original test set. Moreover, we evaluate the fidelity of IFNMTPP using five synthetic datasets. Experiment results on synthetic datasets (reported in Appendix E.2) demonstrate the high fidelity of IFNMTPP compared with other MTPP models.

---

[3]Retweet, StackOverflow, Taobao, and USearthquake are released under Apache-2.0 license[38].
[4]Our codes will be released under MIT license.

## 4.1 Experiment results

**Impact of Thresholding on Mark Prediction**  For the mark prediction, we evaluate (i) the effectiveness of the proposed thresholding method by comparing *ours* with *ours-w/o-thresholding*, (ii) the necessity of the strategy to predict the mark first by comparing *ours* with *time-mark-with-thresholding*. The metric is macro-F1 and micro-F1, where the higher values indicate more accurate mark predictions. The experimental results are reported in Table 2. Compared with *ours-w/o-thresholding*, *ours* performs much better on rare and all mark prediction. *ours* also shows a comparable performance on frequent mark prediction. It implies that the recall of frequent mark prediction is improved using thresholding.

In Table 2, the performance of *ours* is better than *time-mark-with-thresholding* on rare mark prediction on all datasets. To check whether the suboptimal performance of *time-mark-with-thresholding* is due to applying thresholding or not, we also compare *time-mark-with-thresholding* with *time-mark-w/o-thresholding*. The experimental results show that the performance of *time-mark-w/o-thresholding* is lower than that of *time-mark-with-thresholding*. This indicates that predicting the mark first is more suitable for handling mark imbalance with thresholding.

**Time Prediction Performance**  For time prediction performance evaluation, we compare the time predicted using *ours*, i.e., based on $p^*(t|m)$, and that using *time-mark-with-thresholding*, i.e., based on $p^*(t)$. The metric is MAE. Lower MAE means better. The experimental results are reported in Table 3. We observe that predicting time based on $p^*(t|m)$ slightly outperforms that based on $p^*(t)$. The results indicate that the strategy to predict the mark first and then time also benefits the time prediction. The reason could be that $\bar{t}_m$ obtained by drawing samples from $p^*(t|m)$ is more specific to the mark and thus tends to be more accurate compared to $\bar{t}$ obtained based on $p^*(t)$ for all marks.

Table 3: Time prediction performance to evaluate the order of time and mark predictions. The bold are the best values.

| | | Retweet | SO | Taobao | USearthquake |
|---|---|---|---|---|---|
| ours | M | $2515.1_{\pm 6.5029}$ | $\mathbf{0.5212_{\pm 0.0142}}$ | $0.3324_{\pm 0.0579}$ | $\mathbf{0.6856_{\pm 0.0063}}$ |
| | $M_r$ | $3291.2_{\pm 29.097}$ | $\mathbf{0.4986_{\pm 0.0175}}$ | $0.3385_{\pm 0.0627}$ | $\mathbf{0.6966_{\pm 0.0081}}$ |
| | $M_f$ | $2198.7_{\pm 2.4798}$ | $\mathbf{0.6063_{\pm 0.0001}}$ | $0.2529_{\pm 0.0055}$ | $\mathbf{0.6713_{\pm 0.0048}}$ |
| time-mark-with-thresholding | M | $2504.5_{\pm 4.4738}$ | $0.6417_{\pm 0.0127}$ | $\mathbf{0.2420_{\pm 0.0227}}$ | $0.8516_{\pm 0.2378}$ |
| | $M_r$ | $3223.2_{\pm 10.354}$ | $0.6515_{\pm 0.0167}$ | $\mathbf{0.2411_{\pm 0.0215}}$ | $0.7008_{\pm 0.0075}$ |
| | $M_f$ | $2207.7_{\pm 3.1220}$ | $0.6095_{\pm 0.0008}$ | $\mathbf{0.2573_{\pm 0.0429}}$ | $1.2295_{\pm 0.7947}$ |

Table 4: Time prediction performance to evaluate thresholding vs. resampling. The bold are the best.

| | | Retweet | SO | Taobao | USearthquake |
|---|---|---|---|---|---|
| ours | M | $2515.1_{\pm 6.5029}$ | $\mathbf{0.5212_{\pm 0.0142}}$ | $\mathbf{0.3324_{\pm 0.0579}}$ | $\mathbf{0.6856_{\pm 0.0063}}$ |
| | $M_r$ | $3291.2_{\pm 29.097}$ | $\mathbf{0.4986_{\pm 0.0175}}$ | $\mathbf{0.3385_{\pm 0.0627}}$ | $\mathbf{0.6966_{\pm 0.0081}}$ |
| | $M_f$ | $\mathbf{2198.7_{\pm 2.4798}}$ | $\mathbf{0.6063_{\pm 0.0001}}$ | $\mathbf{0.2529_{\pm 0.0055}}$ | $\mathbf{0.6713_{\pm 0.0048}}$ |
| Oversampling | M | $\mathbf{2514.6_{\pm 7.3797}}$ | $6.7145_{\pm 7.8985}$ | $3.2026_{\pm 0.0338}$ | $10.598_{\pm 18.104}$ |
| | $M_r$ | $\mathbf{3197.0_{\pm 6.0093}}$ | $3.2344_{\pm 3.4202}$ | $3.1683_{\pm 0.0375}$ | $10.098_{\pm 18.678}$ |
| | $M_f$ | $2230.1_{\pm 7.7471}$ | $96.441_{\pm 122.71}$ | $3.8056_{\pm 0.0389}$ | $16.139_{\pm 19.098}$ |
| Undersampling | M | $2526.9_{\pm 10.085}$ | $3.5537_{\pm 3.9587}$ | $3.2130_{\pm 0.0313}$ | $17.086_{\pm 21.001}$ |
| | $M_r$ | $3216.9_{\pm 22.171}$ | $1.8360_{\pm 1.4894}$ | $3.1786_{\pm 0.0339}$ | $16.326_{\pm 22.010}$ |
| | $M_f$ | $2239.5_{\pm 5.7513}$ | $54.502_{\pm 76.132}$ | $3.8188_{\pm 0.0367}$ | $26.183_{\pm 18.856}$ |

**Thresholding vs. Resampling**  We compare the prediction performance of *ours* against resampling baselines *oversampling* and *undersampling*. As discussed in Section 1, *resampling the training set* and *cost-sensitive approaches* are two commonly used methods for handling data imbalance besides *thresholding*. According to López et al. [22], *resampling the training set* and *cost-sensitive approaches* are statistically equivalent. So, we focus on *resampling the training set* only. The experiment results are reported in Table 5. We observe that *ours* consistently outperforms *oversampling* and *undersampling*. It is easy to see that the resampling ratio impacts the performance, but it is hard to figure out the correct ratio for different marks on different datasets.

Table 5: Mark prediction performance to evaluate thresholding and resampling, measured by macro-F1/micro-F1, The bold are the best values.

| | | Retweet | SO | Taobao | USearthquake |
|---|---|---|---|---|---|
| ours | M | $\mathbf{0.4750_{\pm 0.0033}/0.4394_{\pm 0.0093}}$ | $\mathbf{0.1776_{\pm 0.0030}/0.6376_{\pm 0.0026}}$ | $\mathbf{0.4190_{\pm 0.0104}/0.7499_{\pm 0.0151}}$ | $\mathbf{0.1382_{\pm 0.0071}/0.3189_{\pm 0.0125}}$ |
| | $M_r$ | $\mathbf{0.2010_{\pm 0.0082}/0.2010_{\pm 0.0082}}$ | $\mathbf{0.1476_{\pm 0.0041}/0.4530_{\pm 0.0026}}$ | $\mathbf{0.3987_{\pm 0.0108}/0.7558_{\pm 0.0185}}$ | $0.0339_{\pm 0.0051}/0.1111_{\pm 0.0098}$ |
| | $M_f$ | $\mathbf{0.6120_{\pm 0.0013}/0.9612_{\pm 0.0021}}$ | $\mathbf{0.2795_{\pm 0.0014}/0.8974_{\pm 0.0042}}$ | $\mathbf{0.7441_{\pm 0.0060}/0.7441_{\pm 0.0060}}$ | $\mathbf{0.2773_{\pm 0.0215}/0.9181_{\pm 0.0102}}$ |
| Oversampling | M | $0.2368_{\pm 0.0197}/0.3484_{\pm 0.0041}$ | $0.0635_{\pm 0.0184}/0.4574_{\pm 0.0496}$ | $0.3538_{\pm 0.0063}/0.7269_{\pm 0.0346}$ | $0.0647_{\pm 0.0165}/0.2141_{\pm 0.0710}$ |
| | $M_r$ | $0.1452_{\pm 0.0016}/0.1452_{\pm 0.0016}$ | $0.0447_{\pm 0.0233}/0.3330_{\pm 0.0340}$ | $0.3341_{\pm 0.0090}/0.8059_{\pm 0.0001}$ | $\mathbf{0.0392_{\pm 0.0071}/0.1818_{\pm 0.0022}}$ |
| | $M_f$ | $0.2859_{\pm 0.0176}/0.2859_{\pm 0.0249}$ | $0.1272_{\pm 0.0001}/0.6284_{\pm 0.0720}$ | $0.6570_{\pm 0.0623}/0.6570_{\pm 0.0623}$ | $0.0988_{\pm 0.0477}/0.2819_{\pm 0.1707}$ |
| Undersampling | M | $0.2284_{\pm 0.0126}/0.3230_{\pm 0.0391}$ | $0.0709_{\pm 0.0226}/0.4171_{\pm 0.0357}$ | $0.3513_{\pm 0.0069}/0.7239_{\pm 0.0102}$ | $0.0576_{\pm 0.0121}/0.3067_{\pm 0.0292}$ |
| | $M_r$ | $0.1422_{\pm 0.0170}/0.1422_{\pm 0.0170}$ | $0.0544_{\pm 0.0263}/0.3017_{\pm 0.0105}$ | $0.3328_{\pm 0.0090}/0.8143_{\pm 0.0048}$ | $0.0382_{\pm 0.0084}/0.1742_{\pm 0.0077}$ |
| | $M_f$ | $0.2714_{\pm 0.0143}/0.7338_{\pm 0.0898}$ | $0.1271_{\pm 0.0174}/0.5850_{\pm 0.1189}$ | $0.6435_{\pm 0.0144}/0.6435_{\pm 0.0144}$ | $0.0836_{\pm 0.0353}/0.5507_{\pm 0.1273}$ |

**Performance Comparison with Existing MTPP models**  Table 7 and Table 6 report time prediction performance and mark prediction performance, respectively, of *ours* and existing MTPP models,

including FullyNN, THP, SAHP, AttNHP, and Marked-LNM. Compared with existing MTPP models, *ours* demonstrates superior performance in both time prediction and mark prediction. For mark prediction, *ours* is the first MTPP model which addresses mark imbalance. For time prediction, the time for each mark $m$ is predicted by drawing samples from $p^*(t|m)$ based on $\Gamma^*(m,t)$. The accurate approximation of $\Gamma^*(m,t)$ leads to accurate time prediction. In particular, *ours* also outperforms Marked-LNM in time prediction. This demonstrates that modeling $\Gamma^*(m,t)$ by neural networks is better than directly modeling $p^*(t|m)$ by the composition of log-normal distributions.

Table 7: Mark prediction performance to evaluate *ours* against existing MTPP models, measured by macro-F1/micro-F1. The bold are the best values.

| | | Retweet | SO | Taobao | USearthquake |
|---|---|---|---|---|---|
| ours | M | **0.4750**±0.0033/**0.4394**±0.0093 | **0.1776**±0.0030/**0.6376**±0.0026 | **0.4190**±0.0104/**0.7499**±0.0151 | **0.1382**±0.0071/**0.3189**±0.0125 |
| | M$_r$ | **0.2010**±0.0082/**0.2010**±0.0082 | **0.1476**±0.0041/**0.4530**±0.0026 | **0.3987**±0.0108/**0.7558**±0.0185 | **0.0339**±0.0051/**0.1111**±0.0098 |
| | M$_f$ | 0.6120±0.0013/0.9612±0.0021 | **0.2795**±0.0014/**0.8974**±0.0042 | **0.7441**±0.0060/**0.7441**±0.0060 | 0.2773±0.0215/0.9181±0.0102 |
| FullyNN | M | 0.2190±0.0000/0.0000±0.0000 | 0.0054±0.0000/0.0000±0.0000 | 0.0094±0.0000/0.0000±0.0000 | 0.0914±0.0000/0.0000±0.0000 |
| | M$_r$ | 0.0000±0.0000/0.0000±0.0000 | 0.0000±0.0000/0.0000±0.0000 | 0.0100±0.0000/0.7209±0.0000 | 0.0000±0.0000/0.0000±0.0000 |
| | M$_f$ | 0.3284±0.0000/0.9768±0.0000 | 0.0236±0.0000/0.9155±0.0000 | 0.0000±0.0000/0.0000±0.0000 | 0.2134±0.0000/**0.9457**±0.0000 |
| SAHP | M | 0.4211±0.0050/0.1540±0.0480 | 0.1134±0.0027/0.5665±0.0059 | 0.0616±0.0327/0.1650±0.1574 | 0.0962±0.0005/0.1237±0.0060 |
| | M$_r$ | 0.0266±0.0135/0.0266±0.0135 | 0.0863±0.0032/0.3500±0.0071 | 0.0269±0.0341/0.0825±0.1009 | 0.0037±0.0010/0.0162±0.0015 |
| | M$_f$ | 0.6183±0.0010/0.9769±0.0001 | 0.2054±0.0011/0.9170±0.0005 | 0.6166±0.0112/0.6166±0.0112 | 0.2196±0.0016/0.9451±0.0002 |
| THP | M | 0.2238±0.0068/0.0000±0.0000 | 0.0859±0.0204/0.3984±0.1867 | 0.0069±0.0035/0.0000±0.0000 | 0.0921±0.0003/0.0000±0.0000 |
| | M$_r$ | 0.0000±0.0000/0.0000±0.0000 | 0.0519±0.0270/0.2120±0.1360 | 0.0074±0.0037/0.7208±0.0001 | 0.0000±0.0000/0.0004±0.0006 |
| | M$_f$ | 0.3357±0.0102/0.9768±0.0000 | 0.2015±0.0025/0.9140±0.0012 | 0.0000±0.0000/0.0000±0.0000 | 0.2149±0.0008/**0.9457**±0.0008 |
| AttNHP | M | 0.4100±0.0049/0.1901±0.0143 | 0.0594±0.0037/0.4548±0.0148 | 0.2930±0.0353/0.6359±0.0415 | 0.1306±0.0041/0.0809±0.0460 |
| | M$_r$ | 0.0373±0.0056/0.0373±0.0056 | 0.0188±0.0000/0.2476±0.0029 | 0.2682±0.0363/0.5868±0.0604 | 0.0012±0.0007/0.0092±0.0079 |
| | M$_f$ | 0.5963±0.0046/0.9747±0.0007 | 0.1972±0.0164/0.8372±0.0643 | 0.6901±0.0189/0.6901±0.0189 | **0.3031**±0.0086/0.9434±0.0012 |
| Marked-LNM | M | 0.4216±0.0021/0.1565±0.0129 | 0.1323±0.0009/0.5995±0.0038 | 0.0911±0.0551/0.6658±0.0615 | 0.1056±0.0048/0.1130±0.0027 |
| | M$_r$ | 0.0252±0.0041/0.0252±0.0041 | 0.1119±0.0001/0.3940±0.0055 | 0.0547±0.0577/0.6278±0.0884 | 0.0063±0.0072/0.0135±0.0006 |
| | M$_f$ | **0.6198**±0.0010/**0.9769**±0.0000 | 0.2016±0.0004/0.9123±0.0011 | 0.7077±0.0308/0.7077±0.0308 | 0.2380±0.0030/0.9451±0.0001 |

# 5  Related Work

**Marked Temporal Point Process** Many MTPP studies specify a separate Conditional Intensity Function (CIF) $\lambda^*(m,t)$ for each categorical mark $m$, based on which $p^*(m,t)$ can be formulated [10, 24, 44, 41, 12, 26, 29]. A more sophisticated intensity function [24, 44, 41, 26] can better capture the system dynamics but will require approximating the integral of $\lambda^*(m,t)$ using a numerical method such as Monte Carlo. Recurrent Marked Temporal Point Process(RMTPP) [11] eludes numerical integral approximation as the CIF and its integral have a closed form, which makes the log-likelihood easy to compute. Recent studies move away from directly modeling CIF. Shchur et al. [33] proposed an intensity-free solution, called Log-NormMix, to infer the density function $p^*(t)$ from a simple distribution such as the mixture of log-normal distributions. Omi et al. [28] proposed FullyNN to model the integral of CIF using a neural network where CIF can be derived by differentiation, an operation computationally

Table 6: Time prediction performance to evaluate *ours* vs. existing MTPP models. The bold are the best values.

| | | Retweet | SO | Taobao | USearthquake |
|---|---|---|---|---|---|
| ours | M | **2515.1**±6.5029 | **0.5212**±0.0142 | 0.3324±0.0579 | **0.6856**±0.0063 |
| | M$_r$ | **3291.2**±29.097 | **0.4986**±0.0175 | 0.3385±0.0627 | **0.6966**±0.0081 |
| | M$_f$ | **2198.7**±2.4798 | **0.6063**±0.0001 | 0.2529±0.0055 | **0.6713**±0.0048 |
| FullyNN | M | 5126.0±854.88 | 0.7047±0.0203 | 6.5079±2.0854 | 1.2684±0.3715 |
| | M$_r$ | 7525.4±1037.4 | 0.7231±0.0269 | 6.6713±2.1557 | 1.2709±0.3356 |
| | M$_f$ | 4232.0±769.50 | 0.6461±0.0029 | 4.4131±1.3632 | 1.2671±0.4082 |
| SAHP | M | 3320.0±242.70 | 0.8010±0.0593 | 23.409±14.564 | 0.7608±0.0588 |
| | M$_r$ | 4260.9±618.21 | 0.7882±0.0734 | 27.638±17.438 | 0.7777±0.0650 |
| | M$_f$ | 2936.3±118.11 | 0.8493±0.0208 | 1.7466±0.8728 | 0.7388±0.0512 |
| THP | M | 3601.1±231.52 | 0.6433±0.0059 | 3.0100±0.2806 | 0.7322±0.0078 |
| | M$_r$ | 4250.6±211.71 | 0.6586±0.0080 | 3.0036±0.2893 | 0.7409±0.0057 |
| | M$_f$ | 3315.2±241.35 | 0.6097±0.0010 | 3.3648±1.1252 | 0.7207±0.0114 |
| AttNHP | M | 3551.1±12.611 | 7.9305±5.9188 | 5.4038±1.3280 | 6.4583±2.2939 |
| | M$_r$ | 4406.5±17.518 | 6.8682±4.8605 | 5.2849±1.2944 | 6.6158±2.3176 |
| | M$_f$ | 3187.8±10.646 | 13.197±11.171 | 7.7158±1.9980 | 6.2544±2.2619 |
| Marked-LNM | M | 2559.8±5.9380 | 0.9067±0.3687 | **0.2058**±0.0079 | 0.7646±0.0026 |
| | M$_r$ | 3314.3±1.2460 | 1.0520±0.5330 | **0.2043**±0.0091 | 0.7773±0.0057 |
| | M$_f$ | 2249.7±7.4050 | 0.6084±0.0007 | **0.2318**±0.0128 | 0.7480±0.0013 |

much easier compared with integration. All MTPP studies discussed so far predict the time of the next event first and then predict the mark. Recently, Waghmare et al. [36] proposes to model $p^*(m)$ using a classifier to predict the mark of the next event and modeling $p^*(t|m)$ to predict the time of the event based on LogNormMix.

Recently, Yuan et al. [40] used a Denoising Diffusion Probabilistic Model (DDPM) to predict the next event in the spatio-temporal point process. Lüdke et al. [23] developed Add-and-thin, a method for modifying event sequences sampled from a Poisson process to match a target distribution by adding or removing events. However, the mark of the spatio-temporal point process is continuous instead of discrete, and Add-and-thin is a temporal point process (TPP) model that does not consider marks. Therefore, these two approaches are out of the scope of our research.

**Imbalanced Data Handling** The techniques for handling imbalanced data, including data-level, algorithm-level, and classifier-level approaches, are designed mainly for improving imbalanced classification tasks. The data-level approach is resampling the training set, including undersampling and oversampling [3]. Most existing resampling methods are based on the Synthetic Minority Over-sampling Technique (SMOTE) algorithm [13, 6, 5]. One benefit of data-level approaches is that they can cooperate with any classifiers. In contrast, algorithm-level approaches are more classifier-specific, such as cost-sensitive methods [18, 21, 9]. The classifier-level method is also known as *thresholding* (or *post-scaling*) which learns thresholds to tune the obtained class probability [19, 7, 8, 35]. The effectiveness of resampling the train set is determined by the resampling ratio, but there is no easy way to figure it out for different classes on different datasets. The cost-sensitive approaches require domain knowledge regarding the importance of different marks to set the cost, but this is not always available [17]. To have a solution with minimum external knowledge and assumptions, this study adopts thresholding.

# 6 Conclusion and Limitation

**Conclusion** It is challenging for existing MTPP methods to accurately predict events of rare marks when the distribution of event marks is highly imbalanced. This is unacceptable in many applications if the rare mark is critical such as major earthquakes. This study introduces the first solution to address mark imbalance in MTPP. Instead of predicting mark based on mark probability directly as in existing studies, we learn thresholds to tune the mark probability normalized by the prior probability to optimize mark prediction. To achieve this goal, this study develops a strategy to predict mark first and then the time by integrating two improper integrations into one and proposing a novel Integration-free Neural Marked Temporal Point Process (IFNMTPP) to approximate the unified improper integration to support time sampling and estimation of mark probability, rather than using computationally expensive numerical improper integration. Extensive experiments on real-world datasets demonstrate the superior performance of our solution against various baselines in the next event mark and time prediction.

**Limitation** As the first effort to address the mark imbalance for MTPP, this study verifies the effectiveness of thresholding, but does not investigate (i) the opportunity to extend the thresholding method to incorporate domain knowledge, such as the importance of rare marks, and (ii) the effectiveness of resampling and cost-sensitive approaches in this situation.

## Broader Impact

This paper presents work whose goal is to advance the field of Machine Learning. Specifically, we want to reveal the mark imbalance to the MTPP community and propose a relatively simple solution to inspire the development of more bias-aware MTPP approaches. There are many potential societal consequences of our work, none of which we feel must be specifically highlighted here.

## Acknowledgement

This research is supported in part by the Australian Research Council (ARC) Discovery Projects DP200101441 and DP210100743.

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

# A The Conditional Joint PDF

This study concerns events with categorical marks. For mark $m$, we define a conditional intensity function $\lambda^*(m, t)$:

$$
\begin{aligned}
\lambda^*(m = k_i, t) &= \lambda(m = k_i, t | \mathcal{H}_t) \\
&= \lim_{\Delta t \to 0} \frac{P(m = k_i, t \in [t, t + \Delta t) | \mathcal{H}_{t-})}{\Delta t} \\
&= \lim_{\Delta t \to 0} \frac{p(m = k_i, t \in [t, t + \Delta t) | \mathcal{H}_{t_l}) \Delta t}{P(\forall j \in \mathbb{N}^+, t_j \notin (t_l, t) | \mathcal{H}_{t_l}) \Delta t} \\
&= \lim_{\Delta t \to 0} \frac{p(m = k_i, t \in [t, t + \Delta t) | \mathcal{H}_{t_l})}{P(\forall j \in \mathbb{N}^+, t_j \notin (t_l, t) | \mathcal{H}_{t_l})} \\
&= \frac{p(m = k_i, t \in [t, t + dt) | \mathcal{H}_{t_l})}{P(\forall j \in \mathbb{N}^+, t_j \notin (t_l, t) | \mathcal{H}_{t_l})}
\end{aligned}
\tag{17}
$$

where $\mathcal{H}_{t_l}$ is the history up to (including) the most recent event, $\mathcal{H}_{t-}$ is the history up to (excluding) the current time, $P(\forall j \in \mathbb{N}^+, t_j \notin (t_l, t) | \mathcal{H}_{t_l})$ represents the probability that no event is observed in time interval $(t_l, t)$ given $\mathcal{H}_{t_l}$.

We denote $P'_m((t_1, t_2) | \mathcal{H}_{t_l})$ for the conditional probability that an event $m$ happens in $(t_1, t_2)$. Following the definition of simple TPP that at most one event happens at every timestamp $t$, the probability that no event occurs in $(t_l, t)$ is:

$$
\begin{aligned}
&P(\forall j \in \mathbb{N}^+, t_j \notin (t_l, t) | \mathcal{H}_{t_l}) \\
=& 1 - \sum_{m \in M} P'_m((t_l, t) | \mathcal{H}_{t_l}) \prod_{n \in M, n \neq m} (1 - P'_n((t_l, t) | \mathcal{H}_{t_l})) \\
=& 1 - \sum_{m \in M} \frac{P'_m((t_l, t) | \mathcal{H}_{t_l})}{1 - P'_m((t_l, t) | \mathcal{H}_{t_l})} \prod_{n \in M} (1 - P'_n((t_l, t) | \mathcal{H}_{t_l})) \\
=& 1 - \sum_{m \in M} F(m, t | \mathcal{H}_{t_l}) = 1 - \sum_{m \in M} F^*(m, t)
\end{aligned}
\tag{18}
$$

where

$$
F^*(m, t) = \frac{P'_m((t_l, t) | \mathcal{H}_{t_l})}{1 - P'_m((t_l, t) | \mathcal{H}_{t_l})} \prod_{n \in M} (1 - P'_n((t_l, t) | \mathcal{H}_{t_l}))
\tag{19}
$$

The conditional joint PDF that the next event is $m$ and occurs in $[t, t + dt)$ is:

$$
p(m = k_i, t \in [t, t + \Delta t) | \mathcal{H}_{t_l}) = \frac{dF^*(m = k_i, t)}{dt}
\tag{20a}
$$

$$
\int_{t_l}^{t} p(m = k_i, t \in [t, t + \Delta t) | \mathcal{H}_{t_l}) d\tau = F^*(m = k_i, t)
\tag{20b}
$$

In this study, $p^*(m, t)$, shorthand of $p(m, t | \mathcal{H}_{t_l})$, is the formal representation of $p(m = k_i, t \in [t, t + \Delta t) | \mathcal{H}_{t_l})$. Note $F^*(m, t)$ in Equation (19) is the probability that only one event happens in interval $[t, t + dt)$ and the mark is $m$. This ensures the MTPP represented by $p^*(m, t)$ is simple. By integrating Equation (20a) and Equation (18) in Equation (17), we have

$$
p^*(m, t) = \lambda^*(m, t)(1 - \sum_{w \in M} F^*(w, t))
\tag{21}
$$

where $\sum_{w \in M} F^*(w, t)$ is calculated from the sum of Equation (17) over marker $m$:

$$
\sum_{w \in M} F^*(w, t) = 1 - \exp(-\int_{t_l}^{t} \sum_{n \in M} \lambda^*(n, \tau) d\tau)
\tag{22}
$$

Then, we solve $p^*(m, t)$:

$$
p^*(m, t) = \lambda^*(m, t) \exp(-\int_{t_l}^{t} \sum_{n \in M} \lambda^*(n, \tau) d\tau)
\tag{23}
$$

which is equivalent with Equation (2).

## B  Proof of the Proposition 3.1

*Proof.* The gradient of IFNMTPP is $-p^*(m, t)$, the function it learns must take the form:

$$\text{IFNMTPP}(m, t) = -\int_{C_1}^t p^*(m, \tau)d\tau + C_2 \tag{24}$$

where $C_1$ and $C_2$ are two constants.

From IFNMTPP architecture, the Integral Estimation Module (IEM) consists of multiple fully connected layers with non-negative weights, and monotonic increasing and unbounded activation functions. Then, it ends with a monotonic decreasing function $\sigma(x) = \frac{1}{1+e^x}$ (as illustrated in Figure 2). Since $\lim_{x \to +\infty} \sigma(x) = 0$, we have:

$$\lim_{t \to +\infty} \text{IFNMTPP}(m, t) = \lim_{t \to +\infty} -\int_{C_1}^t p^*(m, \tau)d\tau + C_2 = 0 \tag{25}$$

Substituting this into the earlier equation, we obtain:

$$C_2 = \lim_{t \to +\infty} \int_{C_1}^t p^*(m, \tau)d\tau = \int_{C_1}^{+\infty} p^*(m, \tau)d\tau \tag{26}$$

Substituting $C_2$ back in Equation (24):

$$\text{IFNMTPP}(m, t) = \int_{C_1}^{+\infty} p^*(m, \tau)d\tau - \int_{C_1}^t p^*(m, \tau)d\tau = \int_t^{+\infty} p^*(m, \tau)d\tau = \Gamma^*(m, t) \tag{27}$$

Thus, the output of IFNMTPP is $\Gamma^*(m, t)$ when its gradient is $-p^*(m, t)$. $\qquad\square$

## C  Technical Details

### C.1  Technical Details about IFNMTPP

In Appendix B, we show that IFNMTPP models $\Gamma^*(m, t)$ when the activation function in the Integral Estimation Module (IEM) is monotonic increasing and unbounded. However, we select `tanh` as the activation function for training stability in the implementation. `tanh` is monotonic but bounded, so $\lim_{x \to +\infty} \text{IFNMTPP}(m, t) = C > 0$, making the implemented IFNMTPP slightly inaccurate. To mitigate this issue, we subtract the original output from the implemented IFNMTPP with $C$. The pseudo code is below:

```python
for layer_idx, layer in enumerate(self.mlp):
    # Hidden status at t for calculating \Gamma^*(m, t)
    output = layer(output)
    output = self.layer_activation(output)

    # Hidden status at t_l for calculating \Gamma^*(m, t_l)
    output_zero = layer(output_zero)
    output_zero = self.layer_activation(output_zero)

    if layer_idx == 0:
        # Hidden status at infinity for calculating \Gamma^*(m, +\infty)
        a.k.a. C.
        output_max = torch.ones_like(output) * self.tanh_parameter
    else:
        output_max = layer(output_max)
        output_max = self.layer_activation(output_max)

probability_integral_from_t_to_inf = self.nonneg_integral(-self.aggregate
    (output))

probability_integral_from_tl_to_inf = self.nonneg_integral(-self.
    aggregate(output_zero))
```

```
21
22  probability_integral_minimal = self.nonneg_integral(-self.aggregate(
        output_max))
23
24  # Shift the output with C when required.
25  if self.removes_tail:
26      regularized_probability_integral_from_t_to_inf = (
        probability_integral_from_t_to_inf - probability_integral_minimal)
27
28      regularized_probability_integral_from_tl_to_inf = (
        probability_integral_from_tl_to_inf - probability_integral_minimal) +
         self.epsilon
29
30  else:
31      regularized_probability_integral_from_t_to_inf =
        probability_integral_from_t_to_inf
32
33      regularized_probability_integral_from_tl_to_inf =
        probability_integral_from_tl_to_inf + self.epsilon
```

## C.2 Technical Details about Thresholding

We use the classic threshold-tuning method [19, 7] to obtain the optimal $\epsilon_m$. Specifically, the method obtains optimal $\epsilon_m$ by taking three steps for each mark $m$. In step 1, we draw the precision-recall curve of $m$. This curve shows us the precision and recall across all possible thresholds. In step 2, since our target is to maximize the F1 score, which is the harmonic mean of the precision and recall, we compute the F1 score for all possible threshold using the precision and recall obtained in the first step. In step 3, the threshold that yields the maximum F1 value is the $\epsilon_m$. The pseudo code is below:

```
1   # For each mark $m \in M$
2   for i in range(num_events):
3
4       # Step 1: draw the precision-recall curve.
5       precision[i], recall[i], thresholds = precision_recall_curve((
        training_results_events_next == i).astype(int),
        scaled_training_results_pm[:, i])
6
7       # Step 2: Calculate the F1 score across all possible threshold.
8       f1s = (2 * precision[i] * recall[i]) / (precision[i] + recall[i])
9       f1s = np.nan_to_num(f1s)
10
11      # Step 3: Pick the threshold that yields the maximal F1 value.
12      ix = np.argmax(f1s)
13      f1[i] = f1s[ix]
14      threshold.append(thresholds[ix])
```

Once $\epsilon_m$ is known for each mark $m \in M$, we predict the next mark as $m$ if $r_m - \epsilon_m$ is maximum. Why? because such $m$ will lead to a higher F1 value compared to other marks.

Please note that we did not train a machine learning model like a neural network to obtain the optimal $\epsilon_m$. Conceptually, one can predict $m$ by selecting $m$ with the maximum $r_m - \epsilon_m$ (i.e., $argmax$) and use the prediction loss to update the parameters of the model producing $\epsilon_m$. However, $argmax$ is non-differentiable so that backpropagation is not allowed to update model parameters. This is why we did not use this method.

## D Experiment Settings

### D.1 Real-world Datasets

We use the following four datasets to evaluate the performance of IFNMTPP.

- *Retweet dataset*[42] records when users Retweet a particular message on Twitter. This dataset distinguishes all users into three different types: (1) normal user, whose followers

count is lower than the median, (2) influence user, whose followers count is higher than the median but lower than the 95th percentile, (3) famous user, whose followers count is higher than the 95th percentile. About 2 million Retweets are recorded, and the average sequence length is 108.This dataset is released under the Apache-2.0 license[38].

- *StackOverflow dataset*(SO)[20] was collected from Stackoverflow[5], a popular question-answering website about various topics. Users providing decent answers will receive different badges as rewards. This dataset collects the timestamps when people obtain 22 badges from the website, and the average sequence length is 72.This dataset is released under the Apache-2.0 license[38].

- *Taobao*[2] records users' interactions on Taobao, an online shopping website from China. These actions include user clicking and buying online items, viewing reviews and comments, or searching for items. The average length of sequences in this dataset is 58, and 17 different marks are available. This dataset is released under the Apache-2.0 license[38].

- *USearthquake*[38] records all earthquakes happened in the continental US from USGS[6]. This dataset has 7 marks, referring to earthquakes with magnitude 2.0 to 2.9, 3.0 to 3.9, 4.0 to 4.9, 5.0 to 5.9, 6.0 to 6.9, 7.0 to 7.9, or 8 and higher. The average sequence length is 16.This dataset is released under the Apache-2.0 license[38].

### D.2 Synthetic Datasets

All synthetic datasets are generated so we do not have any licenses information for them. The code to generate all synthetic datasets comes from the codebase of [28] at https://github.com/omitakahiro/NeuralNetworkPointProcess which is publicly accessible without any licenses.

- *Hawkes process dataset Hawkes_1* was generated utilising Hawkes process:

$$\lambda^*(t) = \mu_0 + \sum_{t_i < t} a \exp(-b(t - t_i)) \tag{28}$$

  where $\mu = 0.2$, $a = 0.8$, and $b = 1.0$.

- *Hawkes process dataset Hawkes_2* was generated utilising Hawkes process:

$$\lambda^*(t) = \mu_0 + \sum_{t_i < t} a_1 \exp(-b_1(t - t_i)) + a_2 \exp(-b_2(t - t_i)) \tag{29}$$

  where $\mu = 0.2$, $a_1 = a_2 = 0.4$, $b_1 = 1.0$, and $b_2 = 20$.

- *Homogeneous Poisson process dataset* was generated using the Homogeneous Poisson process where the conditional intensity function $\lambda^*(t)$ is constant over the entire timeline. This paper assumes $\lambda^*(t) = 1$.

- *Self-correct process dataset* was generated using the temporal point process whose intensity significantly drops when an event happens. The definition of the conditional intensity function is $\lambda^*(t) = \exp(\mu(t - t_i) - \alpha N)$ where $N$ is the number of occurred events, and $\mu$ and $\alpha$ are fixed parameters. In our experiments, we set $\alpha = \mu = 1$.

- *Stationary renewal process dataset* was generated using stationary renewal process, which directly defines the probability distribution over time $p^*(t)$ as a log-normal distribution as shown in Equation (30).

$$p^*(t|\sigma) = \frac{1}{\sigma t \sqrt{2\pi}} \exp(-\frac{\log^2(t)}{2\sigma^2}) \tag{30}$$

  where $\sigma$ is the standard deviation. Here, we set $\sigma = 1$. With Equation (30) and TPP's definition, one could solve the corresponding intensity function by Wolframalpha[7]:

$$\lambda^*(t) = \frac{-0.797885 \exp(-0.5 \log^2(t))}{-t + t \, \mathrm{erf}(0.707107 \log(t))} \tag{31}$$

  where $\mathrm{erf}(x) = \frac{2}{\sqrt{\pi}} \int_0^x \exp(-t^2) dt$.

---

[5]https://StackOverflow.com/

[6]http://earthquake.usgs.gov/earthquakes/eqarchives/year/eqstats.php

[7]https://www.wolframalpha.com

These five synthetic distributions cooperate with a synthetic marking methods. This method generates discrete marks sampled from a uniform distribution. All synthetic datasets have 5 different marks.

### D.3 Metrics

#### D.3.1 Metrics for Synthetic Datasets

For synthetic datasets, the real distribution $\hat{p}^*(m,t)$ is known. We can compare the generated $p^*(m,t)$ against the real one. Most papers report the relative NLL loss, that is, the average of the absolute difference between $-\log\hat{p}^*(m,t)$ and $-\log p^*(m,t)$ on the observed events(if markers are unavailable, $-\log\hat{p}^*(t)$ and $-\log p^*(t)$[28, 33]). The lower relative NLL loss indicates a better performance. However, such a metric only evaluates performance at discrete events, which cannot gauge the overall discrepancy between $\hat{p}^*(m,t)$ and $p^*(m,t)$. So, this paper selects Spearman Coefficient $\rho$ and $L^1$ distance to measure the discrepancy between $\hat{p}^*(m,t)$ and $p^*(m,t)$ over time, while we also report the relative NLL loss for reference.

*Spearman Coefficient* $\rho(X,Y)$ measures the relationship between two arbitrary value sequences, $X$ and $Y$, as defined by Equation (32). If $X$ and $Y$ are more correlated, $\rho(X,Y)$ is higher; lower otherwise. Compared with the Pearson coefficient which is suitable if the relationship between $X$ and $Y$ is linear, Spearman coefficient could better deal with non-linear relationships. Because most probability distributions of TPP are non-linear, we select Spearman coefficient.

$$\rho(X,Y) = \frac{\mathrm{Cov}(\mathrm{Rank}(X),\mathrm{Rank}(Y))}{\sigma_X \sigma_Y} \in [-1,1] \tag{32}$$

where $\sigma_X$ and $\sigma_Y$ are the standard deviations of the values in sequence $X = \{x_1, x_2, \cdots, x_n\}$ and $Y = \{y_1, y_2, \cdots, y_n\}$, respectively. We expect $\rho$ between $\hat{p}^*(m,t)$ and $p^*(m,t)$ is close to 1.

$L^1$ *distance* measures how different two arbitrary functions are in interval $[a,b]$.

$$L^1(f,g) = \int_a^b |f(x) - g(x)| dx \geqslant 0 \tag{33}$$

The smaller the $L^1$ distance is, the more similar $f(x)$ and $g(x)$ are. When $L^1(f,g) = 0$, $f(x)$ almost equals to $g(x)$ in interval $[a,b]$ for any $f(x)$ and $g(x)$, or $f(x) = g(x)$ at every $x \in [a,b]$ if both $f(x)$ and $g(x)$ are continuous.

#### D.3.2 Metrics for Real-World Datasets - macro-F1 & micro-F1

The macro-F1 value and micro-F1 value derives from the F1 value. F1 value has been widely used in almost all binary classification tasks because, compared with accuracy that might be fooled by false positives, F1 value takes accuracy and recall rate in its mind, where the model should correctly mark out positive samples for a better accuracy and negative samples for a better recall rate. The definition of F1 value is:

$$\mathrm{F1} = \frac{2 \times \mathrm{Acc} \times \mathrm{Recall}}{\mathrm{Acc} + \mathrm{Recall}} \tag{34}$$

F1 value is only for the binary classification. Some researchers realise that a multi-class classification can be evaluated by decomposing the original classification task into multiple binary classification tasks and averaging every obtained F1 values. This is how macro-F1 is devised. The expression of macro-F1 is:

$$\mathrm{macro\text{-}F1} = \frac{1}{|M|} \sum_{m=1}^{|M|} \mathrm{F1}_m \tag{35}$$

where $\mathrm{F1}_m$ is the F1 value for marker $m$. macro-F1 treats all classes equally, so it has been widely used in studies addressing class imbalance.

On the other hand, micro-F1 is a global average of F1 values. Specifically, micro-F1 computes the sum of true positives, false negatives, and false positives over all classes then use Equation (34) to obtain the micro-F1. micro-F1 shows the overall performance regardless of the class.

If the mark prediction is based on $p^*(m)$ like our solution, macro-F1 and micro-F1 are independent of time prediction by nature. For baselines where mark prediction is based on $p^*(m|t)$, the mark involved in macro-F1 and micro-F1 is conditioned on the real time of the next event to ensure that macro-F1 and micro-F1 are independent of the time prediction. Specifically, for each real next event $(m = k_i, t')$ in a test set, we compute macro-F1 and micro-F1 using the mark predicted from $p(m|t')$.

### D.3.3 Metrics for Real-World Datasets - Mean Absolute Error (MAE)

The test dataset $T$ contains a subset of real next events. We denote $T_{m=k_i} \subset T$ as those real next events where the mark is $k_i \in M$. The number of events in $T_{m=k_i}$ is $|T_{m=k_i}|$. For each real next event $(m = k_i, t) \in T_{m=k_i}$, we are interested in the evaluation of time prediction. Consider all real next events in $T_{m=k_i}$, $MAE_{m=k_i}$ can be defined:

$$MAE_{m=k_i} = \frac{1}{|T_{m=k_i}|} \sum_{(m=k_i,t) \in T_{m=k_i}} |t - \bar{t}_{m=k_i}| \tag{36}$$

The absolute difference $|t - \bar{t}_{m=k_i}|$, between real time $t$ and the predicted time $\bar{t}_{m=k_i}$ for mark $k_i$, is the prediction error for the real next event $(m = k_i, t)$. Here, $k_i$ is not necessarily the predicted mark so that the time prediction evaluation is independent of mark prediction. $MAE_{M_*}$ is the geometric mean of $MAE_{m=k_i}$ across all marks in $M_*$. $M_*$ can be M, $M_f$, or $M_r$:

$$MAE_{M_*} = \sqrt[|M_*|]{\prod_{k_i \in M_*} MAE_{m=k_i}} \tag{37}$$

where $|M_*|$ is the number of marks in $M_*$.

### D.4 Baselines

#### D.4.1 Group One

The first group of baselines includes: (i) *ours-w/o-thresholding*, which is the same as our method but the mark prediction is not optimized with thresholding. The mark prediction returns the mark with the highest mark probability as described in Section 2.1. The purpose is to evaluate the effectiveness of the thresholding method. (ii) *time-mark-with-thresholding*, that uses IFNMTPP to predict the time of the next event first, and then predicts the mark with the same thresholding method as *ours*. To do that, we predict time $\bar{t}$ which is the mean of $N$ samples from $p^*(t) = \sum_{m \in M} p^*(m, t)$ first, and then modify $p^*(m)$ in Equation (4) to $p^*(m|\bar{t})$ for mark prediction following the procedure described in Section 3.1. The purpose is to evaluate the necessity to predict mark first for handling mark imbalance with thresholding. (iii) *time-mark-w/o-thresholding*, is same as time-mark-with-thresholding but mark prediction is not optimized with thresholding.

#### D.4.2 Group Two

The second group of baselines evaluates thresholding against resampling, another classic technique to address data imbalance, including *undersampling* and *oversampling*. For undersampling, we reduce the frequency of other marks to ensure that they have the same number of training events as the rarest mark. For oversampling, we increase the frequency of other marks so that they have the same number of training events as the most frequent mark. For a fair comparison, the backbone MTPP method is IFNMTPP. After training completes for both baselines, the mark with the highest probability is predicted as the next event mark.

#### D.4.3 Group Three

The third group of baselines includes existing MTPP methods. Since MTPP modeling has been well studied in the past decades, the state-of-the-art methods demonstrate comparable performance. Among them, this study selects the most popular ones as baselines. Four neural MTPP methods based on conditional intensity function (CIF) are FullyNN [28], THP [44], SAHP [41], and AttNHP [26]. Besides these four, another baseline Marked-LNM [36] models $p^*(m)$ using a classifier to predict the mark of the next event and models $p^*(t|m)$ using LogNormMix to predict the time of the event.

- *Fully Neural Network(FullyNN)*[28] uses a neural network to estimate the integral of $\lambda^*(t)$ for the history embedding $\mathbf{h}$ and inter-event time $t$. Then the density function is formulated to predict the time of the next event. We rewrote FullyNN in PyTorch[30] based on the official implementation available at https://github.com/omitakahiro/NeuralNetworkPointProcess, which is publicly accessible without any license.

- *Transformer Hawkes Process(THP)*[44] uses a Transformer-based encoder to represent history as a hidden state **h**. The softplus-based intensity function and the density function are modelled to predict the time of next event. We reproduce this model in PyTorch based on the paper.

- *Self-Attentive Hawkes Process(SAHP)*[41] is based on the same intuition as Continuous-time LSTM(CTLSTM)[24], which generalizes the classical Hawkes process by parameterizing its intensity function with recurrent neural networks. CTLSTM is an interpolated version of the standard LSTM, allowing us to generate outputs in a continuous-time domain. SAHP further improves performance by replacing LSTM with Transformers. Because the only difference between SAHP and CTLSTM is the history encoder, and SAHP has reported achieving better performance than CTLSTM, we only evaluate SAHP in this paper. We reproduce this model in PyTorch based on the paper.

- *AttNHP*[26] is another Transformer-based MTPP model. Different from THP and SAHP, where Transformer only encodes history, and the distribution is extracted from history representations using another deep module, AttNHP merges these two modules into one by directly extracting the distribution from historical events using a Transformer. We use the code provided by the author at `https://github.com/yangalan123/anhp-andtt`.

- *Marked LogNormMix(Marked-LNM)*[36] is an MTPP extension of the LogNormMix[33]. Marked-LNM also follows the MT paradigm by modeling $p^*(m)$ first, then using a composition of log Gaussian distribution to represent $p^*(t|m)$. To the best of our knowledge, Marked-LNM is the only MTPP approach predicting the mark of the next event first and then predicting the time of the event. However, Marked-LNM limits the form of $p^*(t|m)$ as the composition of log Gaussian distributions. This setting introduces inductive biases into the model, which could compromise the model prediction performance. We implement this model in PyTorch by modifying the official LogNormMix code at `https://github.com/shchur/ifl-tpp`. The official codes are released under the MIT license.

### D.5 Data Preprocessing

We prepare synthetic and real-world datasets with normalization. For each dataset, normalization scales the time $t$ of every event in each event sequence by the time mean $\bar{t}$ of all events in all event sequences and standard deviation $\sigma$, as shown in Equation (38):

$$t_{scaled} = \frac{t - \bar{t}}{\sigma} \tag{38}$$

Normalization is useful when the time is relatively large, such as in the Retweet dataset. Table 8 shows how normalization is applied on various datasets.

Table 8: Data preprocessing.

| Dataset | Retweet | StackOverflow | Taobao | USearthquake | five synthetic datasets |
|---|---|---|---|---|---|
| Normalization | ✓ | ✓ | ✓ | ✓ | ✗ |

Our work focuses on predicting when the next event will happen provided a mark, especially a rare mark. For each dataset, we classify if one mark is rare or frequent. The percentages of marks in each dataset are presented in Figure 3. Table 9 shows which marks are classified as frequent and which are classified as rare.

Table 9: Rare marks and frequent marks.

| Dataset name | The number of marks | Rare Mark | Frequent Mark |
|---|---|---|---|
| Retweet | 3 | [2] | [0, 1] |
| StackOverflow | 22 | [1, 2, 6, 7, 9, 10, 11, 12, 13, 14, 15, 16, 17, 18, 19, 20, 21] | [0, 3, 4, 5, 8] |
| Taobao | 17 | [0, 1, 2, 3, 4, 5, 6, 7, 8, 9, 10, 11, 12, 13, 14, 15] | [16] |
| USearthquake | 7 | [3, 4, 5, 6] | [0, 1, 2] |

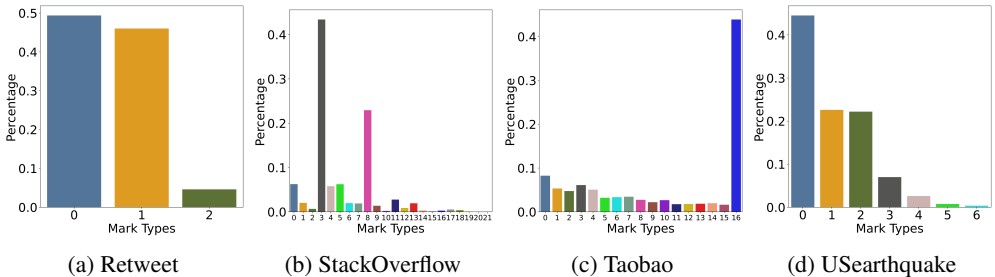

Figure 3: The frequency distribution of marks in real-world datasets.

## D.6 Model Training

This section introduces the hyperparameter settings for all MTPP models used in this paper. The two values of "Steps" refer to the number of warm-up steps and total training steps, respectively. "BS" refers to batch size, and "LR" refers to the learning rate. Unless otherwise specified, we repeatedly train a model 3 times with different random seeds and report the mean and standard deviation of the results. We conduct all experiments on an internal cluster. It includes Intel Xeon CPUs and NVIDIA A100-PCIE GPUs. All codes will be release upon acceptance under the MIT license.

For each mark $m$, we sample $N$ times $\{t^i\}_N^m$ from $F^*(t|m)$ to predict the time of the next event on the condition that its mark is $m$ by the inverse transform sampling:

$$F^*(t^i|m) = u^i \tag{39}$$

where $u^i$ is a random sample from a uniform distribution. The common practice samples $u_i$ from the standard uniform distribution $u^i \sim \mathcal{U}(0,1)$. MTPP allows $t_i$ to go to positive infinity. When $u^i$ is very close to 1, the time drawn from Equation (39) will be meaninglessly big and cause a negative impact to the accuracy of evaluation. To avoid this, we let $u^i \sim \mathcal{U}(0,0.9)$. We find this trick can significantly stabilize the sampling process.

### D.6.1 IFNMTPP Configurations

Table 10 lists the hyperparameter settings for IFNMTPP. The three values of "MS" (model structure) refer to the number of dimensions for history embedding $\mathbf{h}$, the number of dimensions for $\mathbf{v}_m$ and $\mathbf{b}_m$[8], and the number of non-negative fully-connected layers in the IEM module, respectively.

Table 10: Hyperparameter settings for IFNMTPP.

| Datasets | Steps | MS | BS | LR |
|---|---|---|---|---|
| Retweet | [80,000, 400,000] | [32, 16, 4] | 32 | 0.002 |
| Stackoverflow | [40,000, 200,000] | [32, 32, 2] | 32 | 0.002 |
| Taobao | [16,000, 80,000] | [32, 16, 4] | 32 | 0.002 |
| USearthquake | [40,000, 200,000] | [32, 16, 4] | 32 | 0.002 |
| Synthetic | [20,000, 100,000] | [32, 64, 3] | 32 | 0.002 |

### D.6.2 FullyNN Configurations

Table 11 shows hyperparameter settings for FullyNN. The three numbers in column "MS" share the same meaning as those in IFNMTPP.

### D.6.3 THP Configurations

Table 12 shows all hyperparameter settings for THP. The six values of "MS" are the number of dimensions of the Transformer input vectors, the number of dimensions of the hidden outputs from an RNN which is on top of the Transformer encoder, the number of dimensions of the vectors used by self-attentions ($q$, $k$, and $v$), the number of Transformer layers, and heads.

---

[8]$\mathbf{v}_m$ and $\mathbf{b}_m$ always have the same number of dimensions.

Table 11: Hyperparameter settings for FullyNN.

| Datasets | Steps | MS | BS | LR |
|---|---|---|---|---|
| Retweet | [80,000, 400,000] | [32, 16, 4] | 32 | 0.002 |
| Stackoverflow | [40,000, 200,000] | [32, 32, 2] | 32 | 0.002 |
| Taobao | [16,000, 80,000] | [32, 16, 4] | 32 | 0.002 |
| USearthquake | [40,000, 200,000] | [32, 16, 4] | 32 | 0.002 |
| Synthetic | [20,000, 100,000] | [32, 64, 3] | 32 | 0.002 |

Table 12: Hyperparameter settings for THP.

| Datasets | Steps | MS | BS | LR |
|---|---|---|---|---|
| Retweet | [80,000, 400,000] | [16, 16, 32, 8, 3, 3] | 32 | 0.002 |
| Stackoverflow | [40,000, 200,000] | [16, 16, 32, 8, 3, 3] | 32 | 0.002 |
| Taobao | [16,000, 80,000] | [16, 16, 32, 8, 3, 3] | 32 | 0.002 |
| USearthquake | [40,000, 200,000] | [16, 16, 32, 8, 3, 3] | 32 | 0.002 |
| Synthetic | [20,000, 100,000] | [16, 32, 64, 16, 3, 4] | 32 | 0.002 |

### D.6.4 SAHP Configurations

The hyperparameter settings for SAHP are available in Table 13. The first six values of "MS" share the same meaning as those in THP while the last is the dropout rate.

Table 13: Hyperparameter settings for SAHP.

| Datasets | Steps | MS | BS | LR |
|---|---|---|---|---|
| Retweet | [80,000, 400,000] | [16, 16, 32, 8, 3, 3, 0.1] | 32 | 0.002 |
| Stackoverflow | [40,000, 200,000] | [16, 16, 32, 8, 3, 3, 0.1] | 32 | 0.002 |
| Taobao | [16,000, 80,000] | [16, 16, 32, 8, 3, 3, 0.1] | 32 | 0.002 |
| USearthquake | [40,000, 200,000] | [16, 16, 32, 8, 3, 3, 0.1] | 32 | 0.002 |
| Synthetic | [20,000, 100,000] | [16, 32, 64, 16, 3, 4, 0.1] | 32 | 0.002 |

### D.6.5 AttNHP Configurations

The hyperparameter settings for AttNHP are available in Table 14. The first six values of "MS" share the same meaning as those in THP while the last is the dropout rate.

Table 14: Hyperparameter settings for SAHP.

| Datasets | Steps | MS | BS | LR |
|---|---|---|---|---|
| Retweet | [80,000, 400,000] | [16, 16, 64, 8, 3, 3, 0.0] | 32 | 0.002 |
| Stackoverflow | [40,000, 200,000] | [16, 16, 64, 8, 3, 3, 0.0] | 4 | 0.002 |
| Taobao | [16,000, 80,000] | [16, 16, 64, 8, 3, 3, 0.0] | 32 | 0.002 |
| USearthquake | [40,000, 200,000] | [16, 16, 64, 8, 3, 3, 0.0] | 32 | 0.002 |
| Synthetic | [20,000, 100,000] | [16, 16, 64, 8, 3, 3, 0.0] | 32 | 0.002 |

### D.6.6 Marked-LNM Configurations

The hyperparameter settings for Marked-LNM are presented in Table 15. The three values of "MS" are the number of the dimensions of LSTM, the number of the dimensions of mark embedding, and the number of Gaussian distributions, respectively.

## E  Additional Experiment Results

### E.1  Performance of IFNMTPP for modeling $p^*(m, t)$

For a better and integration-free solution, IFNMTPP models the improper integration of $p^*(m, t)$. The advantage has been verified by the experiment results reported in Table 6. IFNMTPP models

Table 15: Hyperparameter settings for Marked-LNM.

| Datasets | Steps | MS | BS | LR |
|---|---|---|---|---|
| Retweet | [80,000, 400,000] | [32, 32, 16] | 32 | 0.002 |
| Stackoverflow | [40,000, 200,000] | [32, 32, 16] | 32 | 0.002 |
| Taobao | [16,000, 80,000] | [32, 32, 16] | 32 | 0.002 |
| USearthquake | [40,000, 200,000] | [32, 32, 16] | 32 | 0.002 |
| Synthetic | [20,000, 100,000] | [32, 32, 16] | 32 | 0.002 |

$p^*(m,t)$ at the same time while modeling the improper integration of $p^*(m,t)$. Compared to other existing MTPP models, the performance of IFNMTPP in modeling $p^*(m,t)$ is evaluated and reported in Table 16. The evaluation metric is NLL loss, the average of the $-\log p^*(m,t)$ at the observed events. The lower NLL loss indicates a better performance. We can observe that IFNMTPP shows a competent performance.

Table 16: Accuracy of $p^*(m,t)$ measured by NLL loss on real-world datasets. Lower is better.

| | IFNMTPP (Ours) | FullyNN | SAHP | THP | AttNHP | Marked-LNM |
|---|---|---|---|---|---|---|
| Retweet | $6.3225_{\pm0.0007}$ | $6.6437_{\pm0.0380}$ | $6.1935_{\pm0.0184}$ | $10.379_{\pm0.5349}$ | $\mathbf{6.0084_{\pm0.0086}}$ | $6.5292_{\pm0.0064}$ |
| Stackoverflow | $\mathbf{2.0540_{\pm0.0029}}$ | $3.6984_{\pm0.0022}$ | $2.0713_{\pm0.0028}$ | $2.5565_{\pm0.0216}$ | $2.0811_{\pm0.0054}$ | $2.0992_{\pm0.0014}$ |
| Taobao | $-0.7762_{\pm0.0565}$ | $-0.0431_{\pm0.0484}$ | $\mathbf{-1.2779_{\pm0.0421}}$ | $140.91_{\pm81.166}$ | $-1.2190_{\pm0.0763}$ | $1.2720_{\pm0.1300}$ |
| USearthquake | $\mathbf{1.3278_{\pm0.0533}}$ | $1.8664_{\pm0.0649}$ | $1.3544_{\pm0.0300}$ | $2.0744_{\pm0.3174}$ | $1.4120_{\pm0.0499}$ | $1.8514_{\pm0.0462}$ |

## E.2 Evaluating Model Fidelity on Synthetic datasets

In this section, we report the full result of model fidelity test on synthetic datasets involving IFNMTPP and other baselines. The IFNMTPP consistently learns more accurate $p^*(m,t)$ than other baselines as supported by the lower $L^1$ distnace and higher Spearman coefficient. These findings suggest that predictions based on IFNMTPP should be more reliable and accurate.

Table 17: Model fidelity test performance on synthetic datasets; higher Spearman, lower $L^1$ and relative NLL loss are better; the bold and underline indicate the best and the second-best values, respectively.

| | | Hawkes_1 | Hawkes_2 | Poisson | Self-correct | Stationary Renewal |
|---|---|---|---|---|---|---|
| Spearman | IFNMTPP (Ours) | $\mathbf{1.0000_{\pm0.0000}}$ | $\mathbf{0.9999_{\pm0.0000}}$ | $\mathbf{1.0000_{\pm0.0000}}$ | $\mathbf{0.9551_{\pm0.0009}}$ | $\mathbf{0.9999_{\pm0.0000}}$ |
| | FullyNN | $0.9952_{\pm0.0004}$ | $0.9963_{\pm0.0002}$ | $0.9722_{\pm0.0018}$ | $0.9477_{\pm0.0001}$ | $\underline{0.9998_{\pm0.0000}}$ |
| | SAHP | $\underline{0.9959_{\pm0.0047}}$ | $0.9862_{\pm0.0000}$ | $0.9615_{\pm0.0025}$ | $0.9492_{\pm0.0014}$ | $0.9990_{\pm0.0007}$ |
| | THP | $0.9266_{\pm0.0026}$ | $0.7366_{\pm0.0005}$ | $\mathbf{1.0000_{\pm0.0000}}$ | $0.6969_{\pm0.0017}$ | $0.0413_{\pm0.0024}$ |
| | AttNHP | - | - | - | - | - |
| | Marked-LNM | $0.9924_{\pm0.0007}$ | $\underline{0.9971_{\pm0.0001}}$ | $0.9713_{\pm0.0024}$ | $\underline{0.9491_{\pm0.0005}}$ | $\mathbf{0.9999_{\pm0.0000}}$ |
| $L^1$ | IFNMTPP (Ours) | $\mathbf{0.1480_{\pm0.0085}}$ | $\mathbf{0.3105_{\pm0.0432}}$ | $\mathbf{0.0133_{\pm0.0091}}$ | $\mathbf{0.5163_{\pm0.0290}}$ | $\underline{0.0654_{\pm0.0018}}$ |
| | FullyNN | $\underline{0.6235_{\pm0.0227}}$ | $3.1048_{\pm0.0763}$ | $0.2973_{\pm0.0098}$ | $1.1889_{\pm0.0244}$ | $0.0710_{\pm0.0099}$ |
| | SAHP | $1.0245_{\pm0.2967}$ | $4.7867_{\pm0.2735}$ | $0.6893_{\pm0.0238}$ | $1.3363_{\pm0.0196}$ | $0.4872_{\pm0.1833}$ |
| | THP | $12.003_{\pm0.2069}$ | $25.500_{\pm0.3642}$ | $\underline{0.0203_{\pm0.0067}}$ | $10.656_{\pm0.0965}$ | $9.9230_{\pm0.0451}$ |
| | AttNHP | - | - | - | - | - |
| | Marked-LNM | $0.6994_{\pm0.0117}$ | $\underline{2.6446_{\pm0.0633}}$ | $0.3620_{\pm0.0044}$ | $\underline{0.7406_{\pm0.0168}}$ | $\mathbf{0.0402_{\pm0.0001}}$ |
| Relative NLL | IFNMTPP (Ours) | $\mathbf{0.0000_{\pm0.0000}}$ | $\mathbf{0.0001_{\pm0.0000}}$ | $\mathbf{0.0000_{\pm0.0000}}$ | $\mathbf{0.0007_{\pm0.0003}}$ | $\mathbf{0.0000_{\pm0.0000}}$ |
| | FullyNN | $\underline{0.0003_{\pm0.0000}}$ | $\underline{0.0008_{\pm0.0001}}$ | $0.0002_{\pm0.0000}$ | $\underline{0.0015_{\pm0.0001}}$ | $\mathbf{0.0000_{\pm0.0000}}$ |
| | SAHP | $0.0086_{\pm0.0017}$ | $0.0312_{\pm0.0193}$ | $0.0092_{\pm0.0002}$ | $0.0072_{\pm0.0009}$ | $\underline{0.0034_{\pm0.0010}}$ |
| | THP | $0.2137_{\pm0.0001}$ | $0.6663_{\pm0.0029}$ | $\mathbf{0.0000_{\pm0.0000}}$ | $0.1262_{\pm0.0004}$ | $0.0771_{\pm0.0000}$ |
| | AttNHP | $0.8202_{\pm0.0053}$ | $0.0387_{\pm0.0144}$ | $0.2631_{\pm0.0009}$ | $0.0820_{\pm0.0003}$ | $0.3065_{\pm0.0007}$ |
| | Marked-LNM | $0.0004_{\pm0.0000}$ | $0.0010_{\pm0.0000}$ | $0.0006_{\pm0.0000}$ | $0.0018_{\pm0.0001}$ | $\mathbf{0.0000_{\pm0.0000}}$ |

### E.3   Comparing IFNMTPP with Marked-LNM and thresholding

Among all baselines, only Marked-LNM follows the mark-time modeling paradigm and is suitable with thresholding. We therefore compare the mark prediction accuracy of our method and Marked-LNM under thresholding. The results, summarized in the table below, demonstrate that our method outperforms Marked-LNM in mark prediction. Note that we do not report time prediction results for this comparison, as the time prediction is unaffected by the method used for predicting marks. Nonetheless, as shown in Table 6, our method also achieves strong performance in time prediction.

Table 18: Comparison of IFNMTPP with Lognormmix + thresholding on four data sets, measured by macro-F1/micro-F1. The bold are the best values.

| | | Retweet | SO | Taobao | USearthquake |
|---|---|---|---|---|---|
| ours | $M$ | **0.4750±0.0033 / 0.4394±0.0093** | **0.1776±0.0030 / 0.6376±0.0026** | **0.4190±0.0104** / 0.7499±0.0151 | **0.1382±0.0071** / 0.3189±0.0125 |
| | $M_r$ | **0.2010±0.0082 / 0.2010±0.0082** | **0.1476±0.0041 / 0.4530±0.0026** | **0.3987±0.0108** / 0.7558±0.0185 | 0.0339±0.0051 / 0.1111±0.0098 |
| | $M_f$ | **0.6120±0.0013 / 0.9612±0.0021** | **0.2795±0.0014 / 0.8974±0.0042** | 0.7441±0.0060 / 0.7441±0.0060 | **0.2773±0.0215 / 0.9181±0.0102** |
| Makred-LNM + thresholding | $M$ | 0.4228±0.0014 / 0.3876±0.0093 | 0.1121±0.0007 / 0.4469±0.0124 | 0.1558±0.0623 / **0.7945±0.0060** | 0.1198±0.0078 / **0.3438±0.0035** |
| | $M_r$ | 0.1730±0.0033 / 0.1730±0.0033 | 0.1004±0.0004 / 0.3527±0.0065 | 0.1181±0.0653 / **0.8318±0.0019** | **0.0393±0.0004 / 0.1740±0.0074** |
| | $M_f$ | 0.5477±0.0004 / 0.8687±0.0005 | 0.1519±0.0017 / 0.5662±0.0209 | **0.7589±0.0133 / 0.7589±0.0133** | 0.2271±0.0169 / 0.6810±0.0427 |

