# OpenReview forum: "Addressing Mark Imbalance in Integration-free Marked Temporal Point Processes"
_NeurIPS.cc/2025/Conference — NeurIPS 2025 poster_

### Official Review · Reviewer_cXr5 · 2025-06-26

**Clarity:** 3
**Significance:** 3
**Originality:** 3
**Rating:** 4
**Confidence:** 4

**Summary:**

The work tackles a practical but under-explored problem—class-imbalance in Marked Temporal Point Processes (MTPPs)—and proposes a neat pipeline that (i) predicts the mark first with per-class F1-optimal thresholds, then (ii) predicts the time using an integration-free neural model that directly approximates a unified tail integral.  Empirical gains on four datasets are convincing, but several methodological assumptions and scalability questions remain unresolved.

**Questions:**

Please refer to the "weaknesses" section above.

**Ethical Concerns:**

["NO or VERY MINOR ethics concerns only"]

**Final Justification:**

Thanks the authors for their comprehensive and thoughtful response to my feedback. I decide to raise my score to 4.

**Limitations:**

Please refer to the "weaknesses" section above.

**Quality:**

3

**Strengths And Weaknesses:**

Strengths
Practical relevance: Rare marks often correspond to critical events (e.g., large quake), yet are poorly handled in existing MTPPs; the paper addresses this gap head-on.
Conceptual simplicity: Thresholding at the probability level leaves the generative time model untouched, avoiding the negative side-effects of resampling or cost-sensitive losses on continuous-time likelihood.
Integration-free inference: The unified-integral trick yields a truly quadrature-free model, leading to faster inference and simpler implementation.
Comprehensive evaluation: Four real-world datasets, ablations on thresholding, mark-first vs. joint prediction, and synthetic studies on integral approximation.

Weaknesses
1.Independence assumption may be too strong. The key factor enabling thresholding is assuming p(m | H_t) is independent of Δt.  Yet many domains exhibit mark–time dependence (e.g., “long silence → rare fault”).  The paper does not measure this dependence nor analyze its impact when violated.

2.Static thresholds under distribution drift. F1-optimal thresholds are learned offline once. In streaming settings where class priors evolve (common in social and medical data), static thresholds can quickly become sub-optimal. No adaptive mechanism is discussed.

3.Scalability to large mark vocabularies. All datasets have ≤ 17 marks.  Threshold search cost is O(|M|·S) and the integral network stores per-class parameters. Feasibility for hundreds or thousands of marks is unclear.

4.Lack of theoretical error bounds. IFNMTPP approximates \Gamma(m,t) with a monotone network, but no approximation error or convergence analysis is provided. It is therefore hard to judge robustness on long horizons or extreme marks.

5.Baseline fairness. The study compares against resampling but omits cost-sensitive loss baselines that are strong for imbalance problems. Without them, the advantage of thresholding may be overstated.

6.End-to-end performance not fully reported. The paper reports MAE conditioned on the true mark and separately macro-F1 for marks. It would be more informative to provide joint metrics (e.g., time MAE using the predicted mark) to quantify error compounding.

---

> ### Author Rebuttal · Authors · 2025-07-31
>
> We appreciate your review.
>
> > Independence assumption may be too strong. The key factor enabling thresholding is assuming $p(m | H_t)$ is independent of Δt. Yet many domains exhibit mark–time dependence (e.g., “long silence → rare fault”). The paper does not measure this dependence nor analyze its impact when violated.
>
> Response:
>
> We respectfully disagree with the claim that our model makes an "independence assumption." While $p^\*(m)$ is the marginal distribution of $p^\*(m, t)$ and thus does not depend on $t$, this alone does not imply that $m$ and $t$ are independent. According to the definition of independence, $m$ and $t$ would be independent only if $p^\*(m, t) = p^\*(m)p^\*(t)$, which is not the case in our model. Instead, we factor the joint distribution as $p^\*(m, t) = p^\*(m)p^\*(t|m)$, as shown in Equations (3) and (6). This formulation preserves the dependence between mark and time via the conditional distribution $p^\*(t|m)$. Therefore, our approach does not assume independence between mark and time.
>
> > Static thresholds under distribution drift. F1-optimal thresholds are learned offline once. In streaming settings where class priors evolve (common in social and medical data), static thresholds can quickly become sub-optimal. No adaptive mechanism is discussed.
>
> Response:
>
> It is true that thresholds may become ineffective if the true data distribution $p^\*(m, t)$ shifts. In our work, $p^\*(m, t)$ is modeled by the MTPP based on historical data. If the MTPP model successfully captures such distributional shifts during training, the predicted probabilities—and thus the derived thresholds—will adapt accordingly. However, if the model fails to capture the shift, the thresholds will remain outdated. Therefore, the adaptability of the thresholds ultimately depends on the MTPP model’s ability to reflect changes in the underlying distribution. In this sense, it is more appropriate to discuss adaptive mechanisms in the context of MTPP modeling rather than thresholding itself, which is beyond the scope of our paper.
>
> > Scalability to large mark vocabularies. All datasets have ≤ 17 marks. Threshold search cost is O(|M|·S) and the integral network stores per-class parameters. Feasibility for hundreds or thousands of marks is unclear.
>
> Response:
>
> We would like to clarify that the StackOverflow dataset in our experiments contains 22 marks.
>
> We agree that scalability to large mark vocabularies is an important concern. Training MTPP models on datasets with hundreds or thousands of marks is a well-known challenge, as it requires computing the integral of $\lambda^\*(m, t)$ for every mark, which incurs high computational cost.
>
> Our proposed IFNMTPP model is integration-free, which avoids this costly computation and makes it more efficient than many intensity-based baselines. However, as noted, the normalization operation across all marks could still become a performance bottleneck when the number of marks is very large. Therefore, while IFNMTPP is well-suited for applications involving dozens of marks—a common setting in many real-world scenarios—scaling to hundreds or thousands of marks remains an open challenge and a promising direction for future work.
>
> > Lack of theoretical error bounds. IFNMTPP approximates $\Gamma(m,t)$ with a monotone network, but no approximation error or convergence analysis is provided. It is therefore hard to judge robustness on long horizons or extreme marks.
>
> Response:
>
> The reviewer may be concerned that the gradient of IFNMTPP is $-p^*(m, t)$, and questions why IFNMTPP is able to learn the cumulative tail probability $\Gamma(m, t) = \int_{t}^{+\infty} p^\*(m, \tau) , d\tau$. We next address this by analyzing how IFNMTPP is trained and what it learns.
>
> We use Maximum Likelihood Estimation (MLE) to train IFNMTPP, a continuous neural network whose gradient with respect to time is $-p^\*(m, t)$. The training loss is the negative log-likelihood (NLL) of $p^\*(m, t)$. Mei et al. (2020) have shown that training with NLL leads to an unbiased MTPP model, meaning the model will learn the true distribution $p^\*(m, t)$ under ideal conditions.
>
> Since IFNMTPP is designed such that its gradient is $-p^\*(m, t)$, the function it learns must take the form:
> $$
> \mathrm{IFNMTPP}(m, t) = -\int_{C_1}^{t}{p^\*(m, \tau)d\tau} + C_2
> $$
> where $C_1$ and $C_2$ are two constants.
>
> From IFNMTPP architecture (lines 181–183), the Integral Estimation Module (IEM) consists of multiple fully connected layers with non-negative weights and monotonic increasing, unbounded activation functions. It terminates with a monotonic decreasing function $\sigma(x) = \frac{1}{1 + e^x}$ for each mark. Since $\lim_{x \to +\infty} \sigma(x) = 0$, and the unbounded activations ensure the IEM’s hidden state diverges as $t \to \infty$, we have:
> $$
> \lim_{t\rightarrow+\infty}{\mathrm{IFNMTPP}(m, t)} = \lim_{t\rightarrow+\infty}{-\int_{C_1}^{t}{p^\*(m, \tau)d\tau} + C_2} = 0
> $$
> Substituting this into the earlier equation, we obtain:
> $$
> C_2 = \lim_{t\rightarrow+\infty}{\int_{C_1}^{t}{p^\*(m, \tau)d\tau}} = \int_{C_1}^{+\infty}{p^\*(m, \tau)d\tau}
> $$
> Substituting $C_2$ back in:
> $$
> \mathrm{IFNMTPP}(m, t) = \int_{C_1}^{+\infty}{p^\*(m, \tau)d\tau}-\int_{C_1}^{t}{p^\*(m, \tau)d\tau} = \int_{t}^{+\infty}{p^\*(m, \tau)d\tau} = \Gamma^\*(m, t)
> $$
> Thus, IFNMTPP always learns the true value of $\Gamma(m, t)$, given that it learns the true density $p^*(m, t)$ through MLE. This theoretical guarantee is a key strength of our method.
>
> Mei, H., Wan, T., & Eisner, J. (2020). Noise-Contrastive Estimation for Multivariate Point Processes. In H. Larochelle, M. Ranzato, R. Hadsell, M. F. Balcan, & H. Lin (Eds.), Advances in Neural Information Processing Systems (Vol. 33, pp. 5204–5214).
>
> > Baseline fairness. The study compares against resampling but omits cost-sensitive loss baselines that are strong for imbalance problems. Without them, the advantage of thresholding may be overstated.
>
> Response:
>
> Section 4.1 (line 254) of our paper explains why cost-sensitive approaches are not included in our comparison. As noted by Lopez et al. [22], resampling methods and cost-sensitive approaches are statistically equivalent. Therefore, by comparing with resampling methods, we effectively cover the evaluation of cost-sensitive strategies as well.
>
> > End-to-end performance not fully reported. The paper reports MAE conditioned on the true mark and separately macro-F1 for marks. It would be more informative to provide joint metrics (e.g., time MAE using the predicted mark) to quantify error compounding.
>
> Response:
>
> When designing our evaluation, we considered the option of measuring "time MAE using the predicted mark." However, we believe this is not a valid evaluation metric. A proper metric must isolate the effect of the variable being. This rule is called *ceteris paribus*, meaning all other influencing factors must be held constant. In the context of MTPP models, the predicted mark directly influences the predicted time through $p^\*(t|m)$. As different models may predict different marks, time errors computed based on these varying predicted marks are not directly comparable. This violates the *ceteris paribus* principle and renders "time MAE using the predicted mark" an invalid metric for evaluating time prediction.
>
> In contrast, "MAE conditioned on the true mark" is a valid and widely accepted metric because the true mark is the same across all models, ensuring fair comparison. This principle also underpins existing evaluation practices in MTPP literature, where time prediction is evaluated via MAE between real and predicted times (from $p^\*(t)$), and mark prediction is assessed via accuracy or NLL conditioned on true times.
>
> In our study, we also use negative log-likelihood (NLL) to evaluate the joint modeling of $p^\*(m, t)$. This provides a unified metric that reflects the model’s performance across both time and mark dimensions simultaneously.

---

> ### Comment · Reviewer_cXr5 · 2025-08-06
>
> Thank you for your comprehensive and thoughtful response to my feedback. I appreciate the thoroughness with which you have addressed each of my concerns, so I decided to raise my score.

---

### Official Review · Reviewer_iCwh · 2025-07-02

**Clarity:** 2
**Significance:** 3
**Originality:** 3
**Rating:** 4
**Confidence:** 4

**Summary:**

This paper proposes a new modeling approach for marked temporal point processes with the intentions of mitigating the impact that imbalance in mark frequencies has on downstream mark prediction. This is done through a combination of two techniques: (i) a proposed MTPP architecture that is integration free through directly parameterizing mark-specific survival functions with monotonic networks, and (ii) by applying a thresholding method when predicting next marks in a sequence to help account for frequency imbalance.

**Questions:**

1. The model proposed probably most resembles the Fully-NN model, which the paper does compare to. I do think the paper does warrant a more in depth comparison in the methodology between these two given their similarities. It is worth noting though that the Fully-NN model was not designed to model marks as it was just a pure TPP model. How then was this model used to compare with in the experiments?
2. If my understanding is correct, the thresholding is applied after the model is initially trained using the log-likelihood objective. From the experiments, it is clear that the thresholding seems to have some positive impact in correcting for mark imbalance. Do you have any potential thoughts on how these lessons could be incorporated into the training of the model directly, to potentially allow for better predictions out of the box without having the transform the distribution post-hoc?

**Ethical Concerns:**

["NO or VERY MINOR ethics concerns only"]

**Final Justification:**

I believe this paper to provide a meaningful contribution towards an under-evaluated topic in MTPPs. While the authors did address the points I raised, I do think the paper would require a good bit of editing to tighten up the messaging and properly showcase their contributions. As such, I will keep my score as-is with hopes of seeing this work accepted, if not here then shortly thereafter in the future after further polishing.

**Limitations:**

yes

**Quality:**

3

**Strengths And Weaknesses:**

**Strengths:**
1. Mark imbalance is a real problem that is common in many datasets, so explicitly targeting this as a problem is well motivated. I also do appreciate that that is the primary focus of the paper as well, compared to say presenting the work as a generic MTPP model (which it admittedly is) and then over index on the next event prediction metrics after the fact.
2. The model presented takes advantage of a clever reformulation of the typical joint distributions that define MTPPs, allowing for direct targets to parameterize to avoid approximating integrals during training.
3. Empirically the method does appear, on average, to perform well and achieves the improvement in next mark prediction that it aimed for.

**Weaknesses:**
1. I felt that the technical details surrounding the proposed model (sections 3.3, 3.4) were a bit lacking in detail and explanation. These sections could have benefited a lot from more text concerning what equations represented and how they pieced together.
2. The empirical results shown established that within the proposed model, thresholding was superior for next mark prediction compared to no thresholding, time then mark prediction (w/ and w/o thresholding), over-sampling, and under-sampling. It also shows that the proposed model with thresholding is in general superior in performance compared to other modeling baselines. Importantly, what it does not show is whether the improvement over baselines is primarily due to the proposed model or thresholding. To have a proper comparison, an additional ablation should be done to apply thresholding (when applicable) to the baseline methods -- at least for some of the experiments as this can understandably be a large undertaking.
3. A minor note, but the empirical results presented in the paper largely consisted of tables of numbers which can be hard to parse at times. Some visualizations of them could go a long way towards improving the readability of the experiments section.

---

> ### Author Rebuttal · Authors · 2025-07-31
>
> We appreciate your review.
>
> > Weaknesses: I felt that the technical details surrounding the proposed model (sections 3.3, 3.4) were a bit lacking in detail and explanation. These sections could have benefited a lot from more text concerning what equations represented and how they pieced together.
> >
> > A minor note, but the empirical results presented in the paper largely consisted of tables of numbers which can be hard to parse at times. Some visualizations of them could go a long way towards improving the readability of the experiments section.
>
> Response:
>
> After carefully reviewing section 3.3 and 3.4, we will add the following content in the final version.
>
> - present the formulas of $F^\*(m|t)$ and $p^\*(m)$ based on $\Gamma^\*(m, t)$ as follows:
> $$
> F^\*(m|t) = \frac{F^\*(m, t)}{p^\*(m)} = \frac{1}{\int_{t_l}^{+\infty}{p^\*(m, \tau)d\tau}}\int_{t_l}^{t}{p^\*(m, \tau)d\tau} = \frac{\Gamma^\*(m, t_l) - \Gamma^\*(m, t)}{\Gamma^\*(m, t_l)}
> $$
> $$
> p^\*(m) = \int_{t_l}^{+\infty}{p^\*(m, \tau)d\tau} = \Gamma^\*(m, t_l)
> $$
>
> - explain the loss function (Equation 12) in details as follows:
>
> The loss function is widely used in MTPP research. For clarity, we present its derivation below:
> $$
> L = -\log p(\mathcal{S}) = -\sum_{(m_i, t_i) \in \mathcal{S}}{\log\lambda^\*(m_i, t_i)} + \int_{0}^{T}{\sum_{n \in M}{\lambda^\*(n, \tau)d\tau}}
> $$
>
> $$
> = - \sum_{(m_i, t_i) \in \mathcal{S}}{(\log\lambda^\*(m_i, t_i)} - \int_{t_{i-1}}^{t_i}{\sum_{n \in M}{\lambda^\*(n, \tau)d\tau})} + \int_{t_l}^{T}{\sum_{n \in M}{\lambda^\*(n, \tau)d\tau})}
> $$
>
> $$
> = - \sum_{(m_i, t_i) \in \mathcal{S}}{\log p^\*(m, t)} - \log(1 - \sum_{n \in M}{F^\*(m, T)})
> $$
>
> $$
> = - \sum_{(m_i, t_i) \in \mathcal{S}}{\log p^\*(m, t)} - \log(\sum_{n \in M}{\Gamma^\*(m, T)})
> $$
> Here, all events occur within the time interval $[0, T]$. The term, $\log(\sum_{n \in M}{\Gamma^\*(m, T)})$, is a log survival term derived from the definition of $p^\*(m,t)$, which represents the conditional density and is the modeling objective in MTPP. This survival term is not a regularization parameter—it cannot be omitted or tuned as a hyperparameter. Excluding it would render the loss function logically incorrect, as it would violate the mathematical formulation of the conditional likelihood in temporal point processes. Therefore, this term is essential for maintaining the theoretical soundness of the model.
>
> > Weaknesses: The empirical results shown established that within the proposed model, thresholding was superior for next mark prediction compared to no thresholding, time then mark prediction (w/ and w/o thresholding), over-sampling, and under-sampling. It also shows that the proposed model with thresholding is in general superior in performance compared to other modeling baselines. Importantly, what it does not show is whether the improvement over baselines is primarily due to the proposed model or thresholding. To have a proper comparison, an additional ablation should be done to apply thresholding (when applicable) to the baseline methods -- at least for some of the experiments as this can understandably be a large undertaking.
>
> Response:
>
> Four of the baselines—FullyNN, SAHP, THP, and AttNHP—are popular intensity-based MTPP models that follow the modeling paradigm of $p^\*(t)$ followed by $p^\*(m|t)$. That is, they first model the event time distribution $p^\*(t)$ to predict the next event time and then predict the mark distribution $p^\*(m|t)$ conditioned on the predicted time. As discussed in Section 1 (paragraph starting at line 40), this paradigm is not suitable for addressing mark imbalance via thresholding, which motivates our alternative modeling paradigm of $p^\*(m)$ followed by $p^\*(t|m)$. This formulation naturally supports thresholding on the mark prediction step and has been empirically validated, as shown in Table 2.
>
> As a result, we do not apply thresholding to the four intensity-based baselines. However, we are still able to compare our method without thresholding (denoted as “ours-w/o-thresholding” in Table 2 for mark prediction and Table 3 for time prediction) against these four baselines (Table 7 for mark prediction and Table 6 time prediction). The results show that our method outperforms the four baselines even without the use of thresholding.
>
> Among all baselines, only Marked-LNM follows the mark-time modeling paradigm and is suitable with thresholding. We therefore compare the mark prediction accuracy of our method and Marked-LNM under thresholding. The results, summarized in the table below, demonstrate that our method outperforms Marked-LNM in mark prediction. Note that we do not report time prediction results for this comparison, as the time prediction is unaffected by the method used for predicting marks. Nonetheless, as shown in Table 6, our method also achieves strong performance in time prediction.
>
> |                     |       | Retweet                        | SO                             | Taobao                          | USearthquake                   |
> |---------------------|-------|--------------------------------|--------------------------------|---------------------------------|--------------------------------|
> | **ours**            | $M$     | **0.4750±0.0033** / **0.4394±0.0093** | **0.1776±0.0030** / **0.6376±0.0026** | **0.4190±0.0104** / 0.7499±0.0151 | **0.1382±0.0071** / 0.3189±0.0125 |
> |                     | $M_r$    | **0.2010±0.0082** / **0.2010±0.0082** | **0.1476±0.0041** / **0.4530±0.0026** | **0.3987±0.0108** / 0.7558±0.0185 | 0.0339±0.0051 / 0.1111±0.0098 |
> |                     | $M_f$    | **0.6120±0.0013** / **0.9612±0.0021** | **0.2795±0.0014** / **0.8974±0.0042** | 0.7441±0.0060 / 0.7441±0.0060 | **0.2773±0.0215** / **0.9181±0.0102** |
> |Lognormmix + thresholding| $M$ | 0.4228±0.0014 / 0.3876±0.0093 | 0.1121±0.0007 / 0.4469±0.0124 | 0.1558±0.0623 / **0.7945±0.0060** | 0.1198±0.0078 / **0.3438±0.0035** |
> | | $M_r$ | 0.1730±0.0033 / 0.1730±0.0033 | 0.1004±0.0004 / 0.3527±0.0065 | 0.1181±0.0653 / **0.8318±0.0019** |  **0.0393±0.0004** / **0.1740±0.0074** |
> | | $M_f$ | 0.5477±0.0004 / 0.8687±0.0005 | 0.1519±0.0017 / 0.5662±0.0209 | **0.7589±0.0133** / **0.7589±0.0133** | 0.2271±0.0169 / 0.6810±0.0427 |
>
> > Q1: The model proposed probably most resembles the Fully-NN model, which the paper does compare to. I do think the paper does warrant a more in depth comparison in the methodology between these two given their similarities. It is worth noting though that the Fully-NN model was not designed to model marks as it was just a pure TPP model. How then was this model used to compare with in the experiments?
>
> Response:
>
> Yes, the original FullyNN is a TPP model that does not support marks. To adapt FullyNN as a baseline for marked TPP models, we modify the input to the "Cumulative Hazard Function Network" (as shown in Figure 1 of the FullyNN paper) by replacing the single time input $\tau$ with a time vector $\tau = [\tau_1, \tau_2, \cdots, \tau_k]$, where each $\tau_i$ corresponds to the time associated with mark $k_i$. Accordingly, the output of the "Cumulative Hazard Function Network" is extended from a scalar to a vector $\Phi = [\Phi(\tau_1), \Phi(\tau_2), \cdots, \Phi(\tau_k)]$. The gradient $d\Phi/d\tau$ is the intensity function for each mark. We will include this description in the final version.
>
> > Q2: If my understanding is correct, the thresholding is applied after the model is initially trained using the log-likelihood objective. From the experiments, it is clear that the thresholding seems to have some positive impact in correcting for mark imbalance. Do you have any potential thoughts on how these lessons could be incorporated into the training of the model directly, to potentially allow for better predictions out of the box without having the transform the distribution post-hoc?
>
> Response:
>
> At present, we do not have an effective approach for training a rare-mark-aware MTPP model that outperforms simple thresholding. However, we consider this an important problem and are actively exploring potential solutions.

---

> > ### Comment · Reviewer_iCwh · 2025-08-06
> >
> > Thank you for the in depth response. This addresses all of my questions.

---

### Official Review · Reviewer_nnTx · 2025-07-03

**Clarity:** 3
**Significance:** 2
**Originality:** 2
**Rating:** 3
**Confidence:** 4

**Summary:**

This work studies the class imbalance problem in MTPP.
The authors propose a novel two-part solution: (1) a thresholding method that adjusts predicted mark probabilities by each mark’s prior frequency to better handle class imbalance, and (2) an Integration-Free Neural MTPP (IFNMTPP) model that predicts the next event’s mark first and then its time without requiring expensive numeric integration.

**Questions:**

- Why not learn thresholds on a validation set instead of the training set to prevent overfitting?
- The Integral Estimation Module (IEM) enforces monotonicity via non-negative weights and activation functions. Does this constraint reduce the flexibility of IFNMTPP in modeling highly non-linear or non-monotonic dynamics?
- The loss includes a log survival term. How sensitive is model performance to this term?

**Ethical Concerns:**

["NO or VERY MINOR ethics concerns only"]

**Limitations:**

1. Thresholding is Post-hoc and Static
2. A mark-specific monotonic neural network (Integral Estimation Module) must be trained for each mark.
3. Applicability Limited to Discrete Marks

**Quality:**

2

**Strengths And Weaknesses:**

Strengths
- It is the first work on class imbalance in MTPPs.
- The proposed thresholding method is an original adaptation of imbalance handling to sequential event prediction.
- the IFNMTPP model introduces a new “integration-free” approach to neural point process modeling.
- The method yields large improvements in rare event prediction (several-fold increases in rare-mark F1 over baselines) while maintaining or improving overall performance.

Weakness
- The IFNMTPP architecture introduces additional complexity (an LSTM history encoder plus a specialized IEM network for each mark) compared to simpler models. Training this model with the monotonic constraints could be challenging too.
- The thresholds for each mark are learned to maximize F1 on the training set. This could potentially overfit those thresholds to training characteristics.
- The study focuses on thresholding vs basic resampling. Other imbalance-handling strategies (e.g., cost-sensitive learning where rare events incur higher loss, or more advanced oversampling like SMOTE variants) are not explored or compared with.

---

> ### Author Rebuttal · Authors · 2025-07-31
>
> We appreciate your review.
>
> > Weakness: The IFNMTPP architecture introduces additional complexity (an LSTM history encoder plus a specialized IEM network for each mark) compared to simpler models. Training this model with the monotonic constraints could be challenging too.
> >
> > Q2: The Integral Estimation Module (IEM) enforces monotonicity via non-negative weights and activation functions. Does this constraint reduce the flexibility of IFNMTPP in modeling highly non-linear or non-monotonic dynamics?
> >
> >Limitation: A mark-specific monotonic neural network (Integral Estimation Module) must be trained for each mark.
>
>
> Response:
>
> First, IFNMTPP does not need a dedicated LSTM history encoder plus a specialized IEM network for each mark, as the reviewer said. As shown in Figure 2, we only have one LSTM providing history information to all marks, and one IEM calculating the value of $\Gamma^\*(m, t)$ for all marks in one forward propagation.
>
> Second, IFNMTPP must be monotonic. Note IFNMTPP is used to model $\Gamma^\*(m, t)$, the integration of the joint distribution $p^\*(m, t)$ from time $t$ to infinity. Clearly, the integration decreases when $t$ increases. So, IFNMTPP must be constrained monotonic.
>
> Finally, the monotocity of IFNMTPP can be easily guaranteed by fully-connected layers of non-negative weights, the monotonic-increasing activation functions, and a monotonically decreasing function $\sigma(x) = 1/(1 + e^x)$, as stated in line 181-183 in section 3.4. The monotonic-increasing activation functions and $\sigma(x)$ are easy to implement. As for fully-connected layers with non-negative weights, we only need to rewrite the ```forward()``` function of the ```nn.Linear``` module as follows:
>
> ```python
>     def forward(self, inputs):
>         weight = self.weight > 0
>         weight = self.weight * weight.float()
>         self.weight.data = torch.clamp(weight, min=self.eps)
>         return F.linear(inputs, self.weight, self.bias)
> ```
> where ```torch.clamp``` will ensure that no negative number exists in the weight of the ```Linear``` module.
>
> > Weakness: The thresholds for each mark are learned to maximize F1 on the training set. This could potentially overfit those thresholds to training characteristics.
> >
> > Q1: Why not learn thresholds on a validation set instead of the training set to prevent overfitting?
>
> Response:
>
> When developing the solution, we were aware that the thresholds fit the training dataset well but may not generalize effectively. However, we would like to emphasize that validation data cannot assist in this case. Typically, during training, a model's performance is evaluated on both training and validation sets. While training loss steadily decreases, validation loss may initially drop and then rise if overfitting occurs. However, as discussed in the last paragraph of Appendix B, we cannot train a model like a neural network to directly optimize thresholds due to the non-differentiable nature of the argmax operation.
>
> Instead, our thresholding method follows [19,7], which determines the optimal threshold for each mark using the training data. Specifically, for each mark $m$, all training events get a probability. These events are then sorted by the probability, and there is a threshold between each pair of consecutive probabilities. For a given threshold, we predict an event as having mark $m$ if its probability exceeds the threshold, and not $m$ otherwise. We compute the F1 score for each possible threshold and select the one yielding the highest F1 as the optimal threshold for that mark.
>
> In theory, one could evaluate different combinations of thresholds across all marks using validation data. However, this approach is impractical for overfitting detection due to (i) the exponential growth in the number of combinations and (ii) the absence of a well-defined analogue to typical model overfitting behavior (i.e., validation loss increasing after an initial decrease), as threshold tuning is a discrete, non-parametric process.
>
> > Weakness: The study focuses on thresholding vs basic resampling. Other imbalance-handling strategies (e.g., cost-sensitive learning where rare events incur higher loss, or more advanced oversampling like SMOTE variants) are not explored or compared with.
>
> Response:
>
> First, Section 4.1 (line 254) of our paper explains why cost-sensitive approaches are not included in our comparison. As noted by Lopez et al. [22], resampling methods and cost-sensitive approaches are statistically equivalent. Therefore, by comparing with resampling methods, we effectively cover the evaluation of cost-sensitive strategies as well.
>
> Second, while SMOTE is a well-known oversampling technique, it is not applicable to our problem. SMOTE works by generating synthetic samples for minority classes to balance the dataset. However, applying SMOTE to a multivariate temporal point process (MTPP) would require an MTPP model capable of generating samples conditioned on a specific mark. To the best of our knowledge, no such MTPP model currently exists. This is why SMOTE is not considered in our study.
>
> > Q3: The loss includes a log survival term. How sensitive is model performance to this term?
>
> Response:
>
> The loss function in Equation (12) is widely used in MTPP research. For clarity, we present its derivation below:
>
> $$
> L = -\log p(\mathcal{S}) = -\sum_{(m_i, t_i) \in \mathcal{S}}{\log\lambda^\*(m_i, t_i)} + \int_{0}^{T}{\sum_{n \in M}{\lambda^\*(n, \tau)d\tau}}
> $$
>
> $$
> = - \sum_{(m_i, t_i) \in \mathcal{S}}{(\log\lambda^\*(m_i, t_i)} - \int_{t_{i-1}}^{t_i}{\sum_{n \in M}{\lambda^\*(n, \tau)d\tau})} + \int_{t_l}^{T}{\sum_{n \in M}{\lambda^\*(n, \tau)d\tau})}
> $$
>
> $$
> = - \sum_{(m_i, t_i) \in \mathcal{S}}{\log p^\*(m, t)} - \log(1 - \sum_{n \in M}{F^\*(m, T)})
> $$
>
> $$
> = - \sum_{(m_i, t_i) \in \mathcal{S}}{\log p^\*(m, t)} - \log(\sum_{n \in M}{\Gamma^\*(m, T)})
> $$
>
> Here, all events occur within the time interval $[0, T]$. The term, $\log(\sum_{n \in M}{\Gamma^\*(m, T)})$, is a log survival term derived from the definition of $p^\*(m,t)$, which represents the conditional density and is the modeling objective in MTPP.
>
> Importantly, this survival term is not a regularization parameter—it cannot be omitted or tuned as a hyperparameter. Excluding it would render the loss function logically incorrect, as it would violate the mathematical formulation of the conditional likelihood in temporal point processes. Therefore, this term is essential for maintaining the theoretical soundness of the model.
>
> > Limitation: Thresholding is Post-hoc and Static
>
> Response:
>
> The reviewer may be concerned that the thresholds might become ineffective if the true distribution $p^\*(m, t)$ of the data shifts. In our work, $p^\*(m, t)$ is modeled by the MTPP based on historical data. If the MTPP model successfully captures such distributional shifts during training, the predicted probabilities—and thus the derived thresholds—will adapt accordingly. However, if the model fails to capture the shift, the thresholds will remain outdated. Therefore, the adaptability of the thresholds ultimately depends on the MTPP model’s ability to reflect changes in the underlying distribution. In this sense, it is more appropriate to discuss adaptive mechanisms in the context of MTPP modeling rather than thresholding itself, which is beyond the scope of our paper.
>
> > Limitation: Applicability Limited to Discrete Marks
>
> Response:
>
> Our problem definition specifically focuses on marked temporal point processes (MTPPs), which model events associated with discrete marks. MTPP with dicrete marks is a well-established framework, with substantial prior research as summarized in Shchur's review [34]. Our paper addresses the rare mark prediction problem within this framework, and therefore, our scope is intentionally limited to discrete marks. We do not consider this a limitation of our work, but rather a deliberate and well-justified modeling choice aligned with the goals of our study.

---

### Official Review · Reviewer_P9hy · 2025-07-07

**Clarity:** 2
**Significance:** 3
**Originality:** 2
**Rating:** 4
**Confidence:** 2

**Summary:**

The paper claims to achieve high performance in discrete imbalanced temporal processes.

**Questions:**

..

**Ethical Concerns:**

["NO or VERY MINOR ethics concerns only"]

**Limitations:**

How do these algorithms scale?

**Paper Formatting Concerns:**

...

**Quality:**

3

**Strengths And Weaknesses:**

First, my understanding is that  the key idea of the paper is to predict separately for the different marks. If I understand right, this by itself does not help much, but makes it possible to generate samples for the rare marks, and then do thresholding.?

The way the authors approach the thresholding is to spend 3.3 to motivate a  complex NN  that includes  a History Encoder, plus a linear stages.  I didn´t quite get it  how you use the LSTM. Would it make sense to  try a regression based approach?

Results:
There are maybe too many results. Namely, always presenting the three configurations makes it difficult to focus on the rare marks, Also, you have different notions of "rare" comparing USearthquake  an say TaoBao. It would be helpful to know basic dataset info such as size and marks frequency.

Macro F1-Micro F1: why these two.  Why not Precision and REcall?

---

> ### Author Rebuttal · Authors · 2025-07-31
>
> We appreciate your review.
>
> > First, my understanding is that the key idea of the paper is to predict separately for the different marks. If I understand right, this by itself does not help much, but makes it possible to generate samples for the rare marks, and then do thresholding?
>
> Response:
>
> We respectively disagree. Our problem aims at predicting the time and mark of the next event, instead of predicting separately for different marks. Specifically, we investigate how to predict events of rare marks more accurately without sacreficing the prediction accuracy on frequent marks. Our solution includes the proposed IFNMTPP and thresholding.
>
> > The way the authors approach the thresholding is to spend 3.3 to motivate a complex NN that includes a History Encoder, plus a linear stages. I didn´t quite get it how you use the LSTM. Would it make sense to try a regression based approach?
>
> Response:
>
> The target function of IFNMTPP $\Gamma^\*(m, t)$  is defined conditional on the history sequence $\mathcal{H}\_{t\_l}$. To properly capture this dependency, we first encode $\mathcal{H}_{t_l}$ into a vector using an LSTM, which is well-suited for processing sequential data with temporal dependencies. This history vector is then fed into the Integral Estimation Module (IEM) of IFNMTPP to model $\Gamma^\*(m, t)$, ensuring that the conditional structure of the problem is preserved.
>
> Given this formulation, we argue that using a standard regression-based approach to model $\Gamma^*(m, t)$ would be inappropriate. Regression methods typically do not account for rich temporal structures or dynamic conditioning on sequences, which are central to the IFNMTPP framework. Therefore, a history-aware encoding followed by conditional modeling is both a principled and necessary design choice.
>
> > Results: There are maybe too many results. Namely, always presenting the three configurations makes it difficult to focus on the rare marks, Also, you have different notions of "rare" comparing USearthquake an say TaoBao. It would be helpful to know basic dataset info such as size and marks frequency.
>
> Response:
>
> First, presenting the macro-F1/micro-F1 for all classes, rare classes, and frequent classes is widely used in research of data imbalanced classification. Following that, we evaluate performance of our model in terms of macro-F1/micro-F1 for all marks, rare marks, and frequent marks in our study. We agree that the result can be a lot, but they comprehensively present the mark prediction performance of our model.
>
> Second, we present the basic dataset info, including dataset size and mark frequency, in Appendix C.5. We place it in the Appendix because we do not have sufficient space in the main paper.
>
> > Macro F1-Micro F1: why these two. Why not Precision and REcall?
>
> F1 value is the harmonic mean of precision and recall, defined as:
> $$
> F1 = \frac{2}{\frac{1}{\mathrm{recall}} + \frac{1}{\mathrm{precision}}}
> $$
>
> F1 value is for binary classification. For multiclass classification, we have Macro F1 and Micro F1. Macro F1 is a macro-averaged F1 score of different marks, while Micro F1 is the harmonic mean of micro precision and micro recall. Macro F1 and Micro F1 have considered precision and recall of different marks, so there is no need to present precision and recall.
>
> > How do these algorithms scale?
>
> The scalability to large mark vocabularies is an important concern. Training MTPP models on datasets with hundreds or thousands of marks is a well-known challenge, as it requires computing the integral of $\lambda^*(m, t)$ for every mark, which incurs high computational cost.
>
> Our proposed IFNMTPP model is integration-free, which avoids this costly computation and makes it more efficient than many intensity-based baselines. However, as noted, the normalization operation across all marks could still become a performance bottleneck when the number of marks is very large. Therefore, while IFNMTPP is well-suited for applications involving dozens of marks—a common setting in many real-world scenarios—scaling to hundreds or thousands of marks remains an open challenge and a promising direction for future work.

---

> > ### Comment · Reviewer_P9hy · 2025-08-04
> >
> > Dear Authors
> >
> > Thanks for your kind answers!

---

### Decision · Program_Chairs · 2025-09-17

**Decision:**

Accept (poster)

**Comment:**

Following the final round of reviews, the paper received one borderline reject and three borderline accept recommendations. The reviewers recognized the significance and novelty of the work, which focuses on developing Marked Temporal Point Process (MTPP) models to address class imbalance, a common issue in real-world datasets. The paper proposes a thresholding method in which class-specific thresholds are learned to maximize the F1 score on the training data. This approach demonstrated strong performance across several MTPP datasets. Reviewers appreciated the empirical results but expressed concerns regarding the sensitivity of the method to the choice of thresholds. They also noted that the proposed architecture introduces additional complexity compared to existing models. The authors’ rebuttal addressed several of these concerns effectively and had a positive influence on the reviewers’ overall assessment. Although some limitations remain, the Area Chair concluded that the paper's strengths outweigh its weaknesses and therefore recommended acceptance.